# Prognostic significance of BIRC7/Livin, Bcl-2, p53, Annexin V, PD-L1, DARC, MSH2 and PMS2 in colorectal cancer treated with FOLFOX chemotherapy with or without aspirin

Mohammed Faruk[1]*, Sani Ibrahim[2], Surajo Mohammed Aminu[3], Ahmed Adamu[4], Adamu Abdullahi[5], Aishatu Maude Suleiman[3], Abdulmumini Hassan Rafindadi[1], Abdullahi Mohammed[1], Yawale Iliyasu[1], John Idoko[1], Rakiya Saidu[6,7], Abdullahi Jibril Randawa[8], Halimatu Sadiya Musa[9], Atara Ntekim[10], Khalid Zahir Shah[11], Sani Abubakar[12], Kasimu Umar Adoke[13], Muhammad Manko[14], Cheh Agustin Awasum[9]

1 Department of Pathology, Faculty of Basic Clinical Sciences, College of Medical Sciences, Ahmadu Bello University, Zaria, Nigeria, 2 Department of Biochemistry, Faculty of Life Sciences, Ahmadu Bello University, Zaria, Nigeria, 3 Department of Haematology and Blood Transfusion, Faculty of Basic Clinical Sciences, College of Medical Sciences, Ahmadu Bello University, Zaria, Nigeria, 4 Department of Surgery, Faculty of Basic Clinical Sciences, College of Medical Sciences, Ahmadu Bello University, Zaria, Nigeria, 5 Department of Radiotherapy and Oncology, College of Medical Sciences, Faculty of Basic Clinical Sciences, Ahmadu Bello University, Zaria, Nigeria, 6 Department of Obstetrics and Gynaecology, Faculty of Medicine, University of Ilorin, Ilorin, Nigeria, 7 Department of Obstetrics and Gynaecology, University of Cape Town, Cape Town, South Africa, 8 Department of Obstetrics and Gynaecology, Faculty of Clinical Sciences, College of Medical Sciences, Ahmadu Bello University, Zaria, Nigeria, 9 Department of Veterinary Surgery and Radiology, Veterinary Teaching Hospital, Ahmadu Bello University Zaria, Zaria, Nigeria, 10 Department of Radiation Oncology, College of Medicine, University of Ibadan, Ibadan, Nigeria, 11 Nova Medical Centre, Lahore, Pakistan, 12 Department of Pathology, Aminu Kano University Teaching Hospital, Kano, Nigeria, 13 Department of Pathology, Federal Medical Centre, Birnin Kebbi, Nigeria, 14 Department of Medicine, Faculty of Clinical Sciences, College of Medical Sciences, Ahmadu Bello University, Zaria, Nigeria

* fmohammed@abu.edu.ng

**Data Availability Statement:** All relevant data are within the paper and its Supporting Information files.

## Abstract

Evasion of apoptosis is associated with treatment resistance and metastasis in colorectal cancer (CRC). Various cellular processes are associated with evasion of apoptosis. These include overexpression of pro-apoptotic proteins (including p53 and PD-L1), anti-apoptotic proteins (BIRC7/Livin and Bcl-2), chemokine receptors (including DARC), and dysregulation of DNA mismatch repair proteins (including MSH2 and PMS2). The aim of this study was to determine the effect of folinic acid, 5-FU and oxaliplatin (FOLFOX) as a single agent and aspirin plus FOLFOX in various combinations on the aforementioned proteins in human CRC, SW480 cell line and rat models of N-Methyl-N-Nitrosourea (NMU)-induced CRC. In addition, effects of the NMU-induced CRC and chemotherapeutic regimens on haematological and biochemical parameters in the rat models were studied. Immunohistochemistry, immunofluorescence and immunoblot techniques were used to study the expression pattern of the related proteins in the human CRC cells pre- and post-treatment. Double contrast barium enema, post-mortem examination and histological analyses were used to confirm tumour growth and the effect of the treatment *in vivo* in rat models. Notably, we found in human mucinous CRC, a significant increase in expression of the BIRC7/Livin post-

**Funding:** The author(s) received no specific funding for this work.

**Competing interests:** The authors have declared that no competing interests exist.

FOLFOX treatment compared with pre-treatment ($p = 0.0001$). This increase provides new insights into the prognostic role of BIRC7/Livin in evasion of apoptosis and facilitation of treatment resistance, local recurrence and metastasis particularly among mucinous CRCs post-FOLFOX chemotherapy. These poor prognostic features in the CRC may be further compounded by the significant suppression of DARC, PD-L1, PMS2 and overexpression of MSH2 and anti-apoptotic Bcl-2 and p53 proteins observed in our study ($p < 0.05$). Importantly, we found a significant reduction in expression of BIRC7/Livin and reactivation of DARC and PD-L1 with a surge in Annexin V expression in rat models of CRC cells post-treatment with a sequential dose of aspirin plus FOLFOX compared with other treatments *in vivo* ($p < 0.05$). The mechanistic rational of these effects underscores the importance of expanded concept of possible aspirin combination therapy with FOLFOX sequentially in future CRC management. Validation of our findings through randomized clinical trials of aspirin plus FOLFOX sequentially in patients with CRC is therefore warranted.

## Introduction

Today, colorectal cancer (CRC) is a significant health problem with a non-uniform increase in incidence, treatment resistance and mortality across the globe [1, 2]. In 2018, CRC remains the third most common cancer and the second most common cause of cancer death worldwide with 1, 800, 977 (10%) newly diagnosed patients and 861, 663 (9%) deaths [2]. However, in sub-Saharan Africa, where cancer registries may lack sufficient resources and there exists a paucity in cancer research, CRC disproportionately presents with advanced stage at diagnosis and a high mortality rate compared to the Western world [2, 3]. These CRC disparities are due in part to issues that include grossly inadequate resources [4] and decreased prioritization of research that focuses on understanding the biology and clinicopathologic features of the disease. These challenges present an urgent need to advance our understanding of CRC in Black Africans in Africa in order to improve prevention, early diagnosis and address treatment resistance for better disease outcomes.

CRC is a disease characterized by the accumulation of genetic and epigenetic alterations resulting in the failure of critical regulatory genes and signalling pathways, which include *p53*, *Bcl-2*, *BIRC7*, *IGF2*, *BRAF*, *PIK3CA*, *WNT*, *PI3K* and *RAS–MAPK* [5, 6]. CRC can be classified as adenocarcinoma not otherwise specified (NOS) and mucinous adenocarcinoma respectively representing 85% and 15% of patients with this malignancy [7]. In addition, four consensus molecular sub-types (CMS) of CRC have been documented, and they include: CMS1 (MSI Immune, 14%) characterized by hypermutation, microsatellite instability (MSI), and strong immune activation; CMS2 (Canonical, 37%) characterized by epithelial, chromosomally unstable features, and clear WNT and MYC signalling activation; CMS3 (Metabolic, 13%) characterized by metabolic dysregulation; and CMS4 (Mesenchymal, 23%) characterised by marked transforming growth factor β activation, stromal disruption and angiogenesis [8]. These classifications of CRC may convey important prognostic information for management of the disease.

Evasion of apoptosis, a hallmark of cancer, plays a pivotal role in sustaining cancer cell growth, survival, treatment resistance, and metastasis in CRC [9, 10]. The overexpression of inhibitor of apoptosis protein (IAP) family including baculoviral IAP repeat-containing protein-7 (BIRC7) have been reported to be contributors of evasion apoptosis in CRC [11, 12].

*BIRC7*, also referred to as *livin* or melanoma inhibitor of apoptosis, has two splice variants termed *Livin α-* and *Livin β-* which, are dissimilar in cellular expression and identical with each other except for a 54 bp truncation at exon 6 [13, 14]. The *BIRC7/Livin* also contains one baculoviral IAP repeat (BIR) and a COOH-terminal RING zinc finger domain composed of seven cysteines and a histidine that coordinate two zinc atoms [15]. Thus, caspase-3–7 and -9 and the second mitochondria-derived activator of caspases (SMAC/DIABLO) are crucial interacting partners of BIRC7/Livin (16). The BIRC7/Livin binds to SMAC/DIABLO via the BIR domain to target the degradation of the SMAC/DIABLO and active caspases [16, 17]. Apoptosis occurs when proteolytic activation of the cysteine-dependent aspartate-directed proteases family (caspases) initiates and cleaves to effector caspases for the breakdown of intra-cellular protein substrates [18]. *BIRC7/Livin* interacts with the SMAC through the BIR domain to promote the caspase activation in the cytochrome c pathway [13, 16]. Notably, dysregulation of the anti-apoptotic protein, B-cell lymphoma-2 (Bcl-2) is also frequently linked to evasion of apoptosis [19]. Here, *Bcl-2* may limit the release of cytochrome c from cellular mitochondria which can result in inefficient apoptosome formation [20]. This dysregulation has been impli-cated in inactivation of the tumor suppressor protein p53 cellular pathway -which plays a cen-tral role in colorectal s pathogenesis [21, 22]. The p53 protein is important for initiation of the apoptotic stimuli during anticancer therapy by sensing DNA damage and activating a series of cellular processes [22]. Thus, under normal cellular conditions, the p53 transcription factor regulates various cell functions such as, apoptosis activation and control of cell growth, migra-tion, and invasion [23]. p53 has also been shown to binds pro-survival Bcl-2 proteins (Bcl-w and Bcl-XL) which release Bax for apoptosis induction [24]. Annexin V, an important marker for detection of apoptotic cells by its ability to bind to phosphatidylserine (outer leaflet of the plasma membrane), has been reported to stimulate immunogenicity of tumour cells [25, 26]. Innate immune cells -such as basophils, dendritic cells, eosinophils, Langerhans cells, mast cells, neutrophils, natural killer cells, monocytes and macrophages and adaptive immune cells which include lymphocytes (B cells and T cells), play important role in the facilitation of effec-tive cancer control via cross-communication [27]. Abnormal function of innate and adaptive immune cells can result to poor cancer cells surveillance, inflammation and evasion of apopto-sis [28, 29]. Therefore, the expression of programmed death ligand-1 (PD-L1), an immune checkpoint marker expressed on tumor cells, facilitates the escape of immunosurveillance in cancer [30]. The program death-1 (PD-1), a cellular protein expressed on the surface of acti-vated immune cells such T cells, NK cells, B cells, macrophages and several subsets of dendritic cells, interacts with PD-L1 to initiate apoptosis of immune cells [31]. This cellular interaction may also be mediated by chemokine receptors such as Duffy antigen receptor for chemokines (DARC)/ atypical chemokine receptor 1 (ACKR1) towards activation of the tumor-specific immune response by chemo-attraction of leukocytes to inflammatory sites and influencing tumor growth and metastasis [32, 33]. Microsatellite instability-high (MSI-H) CRC, a cancer characterized by the absence of one or two mismatch repair (MMR) proteins such as MSH2, PMS2, MLH1, and MSH6 have been reported to be immunogenic with robust lymphocytic infiltrate due to increased mutational signatures [34] and this may be more frequent in CRC with mucinous histology. This suggests that MSI-H CRCs may uniquely benefit from immu-nomodulatory drugs, such as PD-1 inhibitors [34].

Evidence supports the effectiveness of aspirin in inhibition of CRC growth and also in reducing the risk of CRC [35, 36]. Aspirin inhibits cyclooxygenase (COX), reducing PGE2 pro-duction and inducing a pro-tumour inflammatory profile -aspirin may revert this towards an important anti-cancer immune pathway and possibly serve as an adjuvant for immune check-point therapy [36, 37]. Importantly, combined chemotherapeutic agents, -such as folinic acid, 5-FU and oxaliplatin (FOLFOX) have been shown to be effective for CRC management [38].

However, treatment resistance in some CRC patients managed with FOLFOX limits the drug efficacy and may lead to disease recurrences, metastatic spread and poorer prognosis. Thus, there is a need for additional strategies to overcome treatment resistance and treatment-associated serious adverse events in CRC. In this study, we used a systemic approach to elucidate the expression pattern and prognostic significance of Pro and Anti-BIRC7/Livin, Bcl-2, p53, Annexin V, PD-L1 and DARC, and DNA MMR proteins MSH2 and PMS2 in human CRC cells pre- and post-FOLFOX treatments from patients at a West African health institution (Ahmadu Bello University Teaching Hospital) in Zaria Nigeria. We then explore expression patterns of BIRC7/Livin and survival probability among African-American, White and Asian patients with CRC from the Cancer Genome Atlas (TCGA) using UALCAN domain [39]. Next, via immunoblotting, we quantified the expression of BIRC7/Livin and the effect of 0.5 mM aspirin on BIRC7/Livin in the protein proficient, p53 mutant SW480 CRC cell line. We then induced CRC in rats using N-Methyl-N-Nitrosourea (NMU) to further expand our understanding of CRC biology. We also studied the efficacy and *in vivo* effects of aspirin with or without neoadjuvant FOLFOX on the aforementioned proteins in rats. In addition, biochemical and hematologic parameters in NMU model of the CRC in rats were analyzed. Our findings provide a mechanistic understanding and the clinical relevance of aspirin plus FOLFOX combination therapy in sequence as a novel anticancer treatment regimen for CRC.

## Materials and methods

### Multi-racial comparison of BIRC7/Livin expression and survival probability in colorectal cancer

Using a domain that facilitates the analysis of survival and expression of different genes from Cancer Genome Atlas (TCGA), UALCAN, [39], we studied the following: expression pattern of BIRC7/Livin based on patient race; effect of BIRC7/Livin expression level on CRC survival and; effect of BIRC7/Livin expression and race on CRC survival. Race was classified as: African-American, Whites and Asian.

### Cell line, cell culture and treatment *in-vitro*

The human CRC-derived SW480 cell line from the European Collection of Cell Culture (ECACC, Health Protection Agency Wiltshire, UK) was used. The master cell bank was expanded upon arrival and a new ampule of the cell was thawed on 4-months bases for further experiment. The cells were cultured *in vitro* in 75 cm$^2$ flasks, using L-15 medium containing 10% fetal bovine serum (FBS) (PAA Laboratories Ltd, Somerset, UK), supplemented with 1% (w/v) L-glutamine-penicillin-streptomycin (Sigma). The cells were maintained at 37˚C in a humidified incubator with 5% $CO_2$ and regular passage at approximately 80% confluency. For *in-vitro* SW480 cell treatment, aspirin was dissolved in dimethyl sulfoxide (DMSO) solution (Sigma: #BCBD7455V) to a stock concentration of 0.5 mM. The cellular extracts were treated with the 0.5 mM aspirin for 24 hours and a portion termed non-aspirin treated cells were treated with DMSO alone as a vehicle control.

### Western blots

Whole-cell lysates of the SW480 cell line were used for the western blots analysis (50 μg protein per lane) as described as follows: Primary antibody BIRC7/Livin (1:1000; Cell Signalling Technology, #5471P), and a secondary Anti Rabbit IgG HRP-linked antibody (1:1000; Cell Signalling Technology, #7074P2) were used for the immunoblot analysis. In detail, the CRC cells (treated with DMSO or with 0.5 mM aspirin) were repeatedly washed with

phosphate buffered saline (PBS), then protein was extracted using protein extraction buffer (20 mM Tris/ 150 mM NaCl/1 mM EDTA/1% Triton X) on ice for 1 hour, and centrifuged at 13000 g to remove cell fragments. Extracted cellular proteins were dissolved in Laemmli 2X concentrate sample buffer (Sigma, 04M6092) and analysed by discontinuous 15% SDS-PAGE transferred to a membrane and probed for protein expression. Resolved proteins were transferred onto nitrocellulose membrane (Bio-Rad) at 10 v overnight or at 100 v for 1 hour through wet-transfer (Bio-Rad Laboratories). The nitrocellulose was blocked with 5% (w/v) non-fat milk/0.2% (v/v) Tween-20 in Tris-buffered saline (TBS) for 60 minute at room temperature and incubated with the primary antibody overnight at 4˚C, prior to repeated rinsing with 0.2% (v/v) Tween-20 in PBS, and then incubated with a secondary HRP conjugated antibody (Anti Rabbit IgG HRP-linked antibody, Cell Signalling Technology, #7074P2) for 60 minute at room temperature. This was repeatedly rinsed with 0.2% (v/v) Tween-20 in PBS and visualised using ECL reagent (Pierce ECL2 solution A, #1896433A; Pierce ECL solution B, #1896433B) in accordance with manufacturer's instructions using a STORM Phosphor-imager (GE Healthcare UK Limited, Buckinghamshire, UK). To illustrate equal band loading, the samples were probed for PCNA protein expression using a monoclonal anti-mouse HRP-linked antibody.

## Rats

Male albino rats aged seven weeks were purchased from animal facility of the Faculty of Pharmaceutical Sciences, Ahmadu Bello University (ABU) Zaria, Nigeria, under pathogen-free conditions. The rats were accommodated in polypropylene cages, 4 per cage, in standard laboratory conditions with *ad libitum* access to food and water for acclimatization up to the time they attained the weight of 130-175g for cancer induction study and throughout the study period. To minimize the rats suffering and distress, we worked as a team with veterinarians and animal care personnel from the ethic committee and staff members who are experienced in handling laboratory animals at all stage of the experiment and throughout the study period for appropriate monitoring and guidance. Rats were kept at 25 ± 1˚C and 45–55% relative humidity with a 12 hours light/dark cycle in the animal facility of the Faculty of Pharmaceutical Sciences, ABU Zaria, Nigeria, and monitored daily for complete duration of the study. They were labelled and weighed at regular time interval throughout the study period. The study protocol was reviewed and approved by the ABU committee on Animal use and care, Directorate of Academic Planning and Monitoring, ABU Zaria Nigeria (Approval number: ABUCACU/2016/008). The ethics committee at ABU (ABU committee on Animal use and care- https://abu.edu.ng/animal-use/) who reviewed and approved this study contains animal welfare experts from Faculty of Veterinary Medicine of the University.

## Rat grouping and cancer induction

The rats (N = 80) were grouped into 6 sets of 10 rats each and a three-staged study was performed. Group 1 rats were given 0.5 ml of water during CRC induction. Group 2–8 were treated every other day for 60 days with intra-rectal dose of 2 mg/ kg N-Methyl-N-Nitrosourea (NMU; Shijiazhuang Aopharm Medical Technology Co., Ltd, China #684-93-5) dissolved in distilled water to induce CRC. Briefly, a metal feeding tube that is 8 cm long was inserted two-thirds of the way into the colon through the anal orifice, and the NMU solution was infused. The solution was made to fill the distal half of the colon, where the CRC is expected to develop. The CRC induction was confirmed by barium enema and histology analysis.

## Double contrast barium enema

The barium enema procedure involved preparing the entire bowel by infusing about 2 mls of barium solution (Sanochemia Dg, Deutschland) intra rectally before X-ray analysis. The X-ray procedure was performed at the Department of Veterinary Surgery and Radiology, ABU Veterinary Teaching Hospital Zaria, Nigeria, using MDX-100 X-ray machine (Recorders and Medicare Systems Ltd, India).

## Aspirin and chemotherapy treatments procedure *in vivo* in rat

Aspirin was administered at a dose of 25 mg/kg (full) or 12.5 mg/kg (half) daily and orally. The rats were regrouped (S1 Appendix) to receive the following treatments: Group 1 and 2 (Vehicle controls) received intra-peritoneum (IP) treatment of normal saline (0.9% w/v; JUHEL Nigeria Ltd, #21E18). Group 3 received 25 mg/kg aspirin (Vasoprin Ltd®; JUHEL Nigeria, #04–1797) or 25 mg/kg analytical aspirin (Shijiazhuang Aopharm Medical Technology Co., Ltd, China) for 2 days orally. Group 4 received 50 mg/kg 5-FU (Celon Laboratories LTD India, #FUI1504BC) IP, in normal saline alone (0.9% w/v; JUHEL Nigeria Ltd, #21E18)] for 5 days. Group 5 received 25 mg/kg aspirin orally for 2 days and followed by 7 mg/kg folinic acid (Leucovorin-TEVA; TEVA Pharmaceutical Industries Ltd Macaristan, #945041401) in normal saline IP, 3mg/kg oxaliplatin (Miracalus Pharma Ltd Mumbai, India, NN4391B) in dextrose 5% w/v (Dana Pharmaceutical Ltd, Nigeria, #10128) IP and 50 mg/kg 5-FU in normal saline for 5 days IP sequentially. Group 6 received 25 mg/kg aspirin orally, 7 mg/kg folinic acid in normal saline 0.9% w/v IP, 3 mg/kg oxaliplatin in dextrose 5% w/v (Dana Pharmaceutical Ltd, 10128) IP and 50 mg/kg 5-FU in normal saline for 5 days IP concurrently. Group 7 received 7mg/kg folinic acid alone for 5 days. Group 8 received 3 mg/kg oxaliplatin alone for 5 days.

## Post-mortem examination for animal study

Post-mortem (PM) examination was conducted on the rats upon completion of treatment. The rats were sacrificed after tumour confirmation and treatment using Nembutal anaesthesia with xylazine (16 mg/kg) and ketamine (120 mg/kg) before the PM study. The colon, rectum and any other area with histological changes were removed and immediately fixed in 10% formalin for histology analysis.

## Histology analysis of the rat tissues

The entire bowel and other organs with areas of histologic interest obtained from the PM were fixed in 10% formalin for 24 hours before surgical cut up. The sections of interest were sampled, processed and stained with haematoxylin and eosin (H & E), immunohistochemistry (IHC) and immunofluorescence (IF) to examine pro-apoptotic and anti-apoptotic proteins and DNA MMR proteins expression as described in the section below. An abnormal kidney structure observed in one of the rats was histologically processed and stained with Anti-PD-L1 polyclonal for PD-L1 protein expression study.

## Immunohistochemistry and immunofluorescence studies

The IHC and IF were performed on 5-μm FFPE tissue sections. For IHC, the expression of antigens were evaluated with Anti-BIRC7 polyclonal antibodies from Antibodies-online (Aachen, Germany; ABIN358607; 1:80 dilution and ABIN672561; 1:100), Anti-Annexin V polyclonal antibody from Antibodies-online (Aachen Germany; ABIN4964891; 1:80 dilution), Anti-PD-L1 (CD274) monoclonal antibody from Antibodies-online (Aachen,

Germany; ABIN5027498; 5 μg/mL), Anti-DARC polyclonal antibody from Antibodies-online (Aachen Germany; ABIN2821184; 1:50), Anti-MSH2 polyclonal antibody from Antibodies-online (Aachen Germany; ABIN3185692; 1:100), Anti-PMS2 polyclonal antibody from Antibodies-online (Aachen Germany; ABIN5546942; 1;30), Anti-Bcl-2 monoclonal antibody from Genemed Biotechnologies, Inc. (CA, USA; Clone Bcl-2-100; 1:60 dilution) and Anti-p53 monoclonal antibody from Genemed Biotechnologies, Inc. (CA, USA; Clone BP-53-12; 1:60 dilution) to demonstrate BIRC7, Annexin V, PD-L1, DARC, MSH2, PMS2, Bcl-2 and p53 proteins expression. Briefly, the slides were baked for 1 hour, then deparaffinized with 100% xylene at room temperature for 1 minute, and hydrated in graded alcohol stages consisting of 30-second dips each in 100% and 95% ethyl alcohol diluted in water at room temperature, and finally hydrated in water. Sections were incubated in 3% hydrogen peroxidase in water at room temperature for 10 minutes to block endogenous peroxidase activity. The slides were then washed, blocked and incubated at room temperature for 30 minutes. The slides were incubated with the primary antibodies in 1:80 dilutions and with HRP-linked secondary antibody in blocking buffer. The nuclei of the cells were counterstained with haematoxylin (blue). Expression levels were categorized as low and high based on a combined score of intensity and distribution of each protein. Proteins expression was categorized based staining intensity (0 = absent; 1 = weak; 2 = moderate; 3 = strong) and distribution (per cent of tumour positive for the proteins). For IF, sections were further blocked with 1% bovine serum albumin (BSA) and 5% Goat serum for 30 minutes and incubated with Anti-BIRC7 antibody from Antibodies-online (Aachen, Germany, ABIN672561; 1:100 dilution) for 1 hr at room temperature. Goat Anti-Rabit IgG, heavy and light chain antibody from Antibodies-online (Aachen Germany; ABIN101988) were used as secondary antibody.

## Hemogram analysis

Blood samples were collected via venepuncture of abdominal aorta into ethylenediamine-tetraacetic acid (EDTA) bottle for the haematological parameters study in rats. The collection of the blood was done by animal care expert under the supervision of senior veterinarian while restraining the rat manually and minimising the time of restrain and amount of blood collected so as to reduce stress and pain to the rats. The complete blood count for the study of red blood cells (RBC), haemoglobin, haematocrit, platelet, total white blood cells (WBC) and differentials (Lymphocytes and Granulocytes) was carried out using α-Swelab machine (#24403, Sweden). The analysis was carried out immediately after blood sample collection.

## Biochemical study

The blood samples for biochemical analysis were collected into plain sample tubes and allowed to stand for 30 min before centrifugation. Serum was transferred into clean plain sample bottles and then analyzed for creatinine, urea, sodium, potassium, chloride, bicarbonate, aspartate aminotransferase, alanine aminotransferase and alkaline phosphatase immediately. Serum urea and creatinine were estimated by modified diacetyl monoxime and Jaffe kinetic methods. Sodium and potassium were estimated using flame photometer, chloride using Schales and Schales titration method whereas bicarbonate was estimated using the Scribner-Caillouette method. Serum aspartate aminotransferase and alanine aminotransferase were estimated using the colourimetric method of Reitman and Frankel whereas ALP was estimated using King and Armstrong procedure.

## Human colorectal cancer cells and chemotherapeutic regimen

Formalin-fixed, paraffin-embedded (FFPE) CRC tissue samples from 92 patients were collected from the Zaria institutional histology tissue bank. The series included 23 mucinous CRC and 69 CRC NOS FFPE tissue blocks selected from January 2009 to December 2017. Among these cases, 16 patients received neoadjuvant FOLFOX chemotherapy from September 2010 to December 2013 after a biopsy-confirmed histological diagnosis of CRC. The chemotherapy treatment modality used for these patients consisted of 3 or 4 courses of folinic acid 200 mg/m$^2$ (or 100mg/m$^2$) i.v (2 hour infusion) days 1 and 2, Oxaliplatin 85mg/m$^2$ i.v (2 hour infusion) day 1 concurrent then 5-FU 400mg/m$^2$ i.v bolus on days 1 and 2 and 5-FU 600 mg/m$^2$ i.v (22 hour infusion) days 1 and 2. The patients had a complete surgical resection after neoadjuvant chemotherapy followed by 2 or 3 courses as adjuvant treatment. Also, cancer free colorectal tissues were used in this FFPE study as control. Patients demographic data, including age and sex, were collected from the Zaria cancer registry (2010–2017). The human study protocol was reviewed and approved (MOH/ADM/774/VOL1/340) by the institutional Health Research Ethics Committee in the Ministry of Health and Human Services of Kaduna State Government, Nigeria.

## Statistical analysis

Statistical analysis was carried out using the SPSS software, version 20.0 (IBM Corp., Armonk, NY, USA). The mean and standard deviation of the protein expressions were compared by using independent sample t -test. A P -value of <0.05 was considered statistically significant. The bar charts and dot plots were produced by utilizing Microsoft Excel (Microsoft Office Professional Plus 2013; Microsoft Corp., Redmond, WA, USA) to produce simple clustered column charts and dot plots. ImageJ software version 1.4.3.67 (broken Symmetry Software) was used to analyze the H-Score of the BIRC7 protein expression using the following formula:

$$\text{Optical Density (OD)} = \log (\text{maximum intensity/Mean intensity})$$

## Results

### Suppression of BIRC7/Livin by aspirin with or without FOLFOX facilitates apoptotic cell death in colorectal cancer

BIRC7/Livin is an important IAP protein associated with tumor progression, epithelial mesenchymal transition and metastasis via inhibition of apoptosis and autophagy in CRC [40, 41]. Firstly, we used the UALCAN domain [39] to compare expression pattern of BIRC7/Livin and survival among African-Americans, Whites and Asians from Cancer Genome Atlas (TCGA) to explore the importance of BIRC7/Livin expression in CRC among ethnic groups. Results show a 3-fold increase of BIRC7/Livin expression in CRC from Whites (N = 193) compared with African-Americans (N = 55), and about a 2-fold increase in Asians (N = 11) compared with the African-Americans (Fig 1A). There were no significant difference when cumulatively comparing the high and low-medium expression of BIRC7 with patient survival for a 14 months period (Fig 1B, $p$ = 0.056). The results also showed no significant difference when comparing BIRC7/Livin expression with survival among African-Americans, Whites and Asians (Fig 1C, $p$ = 0.49). However, there was limited sample size among minorities in TCGA such that larger sample sizes may reveal clinically relevant differences in expression of BIRC7 for disease outcomes.

Secondly, since the *in vitro* cell culture model has been a crucial tool for hypothesis-driven cancer research, we probed the expression of BIRC7/Livin protein in the SW480 CRC cell line by immunoblot. Results showed an increased expression of the BIRC7 in the SW480 cells *in-*

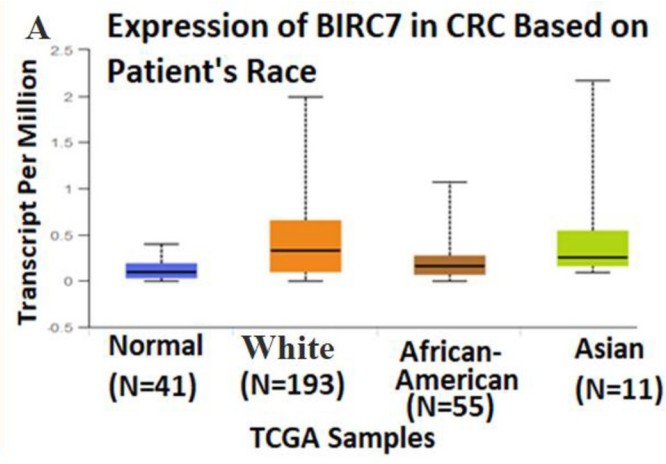

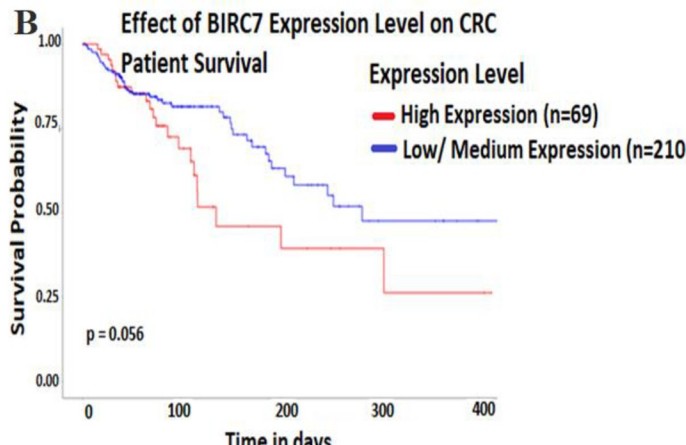

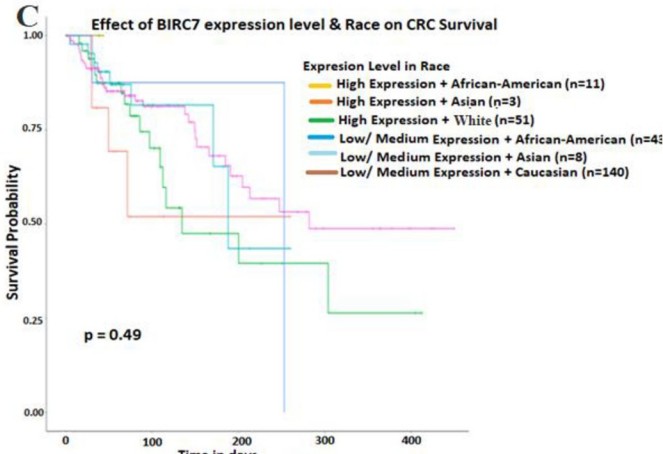

**Fig 1. BIRC7/Livin expression pattern and survival probability in African-American compared with White and Asian race. A,** Representative expression of BIRC7/Livin in CRC based on patient race. **B,** Representative effect of BIRC7/Livin expression level on CRC patient survival. **C,** Representative effect of BIRC7/Livin expression level and race on CRC patient survival.

*vitro* (Fig 2A). The subcellular fraction of the SW480 cell line using Anti-BIRC7/Livin demonstrated clear cytoplasmic and nuclear suppression of the BIRC7 post-treatment with 0.5 mM aspirin for 24 hours compared with DMSO treated cells (Fig 2A and 2B). We then examined the effects of 0.5 mM dose aspirin on the cells. Results of the aspirin treatment revealed reduced SW480 cellular growth compared with DMSO control treatment (Fig 1C). The proteins and functional pathways implicated in aspirin-induced CRC cells death in the SW480 cells are depicted in Fig 2D below.

The third part of this study sought to evaluate expression patterns of BIRC7/Livin in human CRC tissues using IHC and IF protocols. This includes data from ninety two (92) rural African CRC patients with male to female ratio of 2:1 and mean age of 43.1±16.0 (Table 1). Twenty three (23) of the patients were diagnosed with mucinous CRC and sixty nine (69) were CRC not otherwise specified (NOS). About 78% of the patients in this study were below the age of 59 years (Fig 3A). The patients were either treated or untreated with neoadjuvant FOLFOX regimen. Results show significant expression of BIRC7/Livin in CRC tissues compared with normal colorectal sections (Fig 3B–3F and 3H; *p* = 0.0001). The expression patterns of BIRC7/Livin was significantly higher in CRC not otherwise specified compared with mucinous

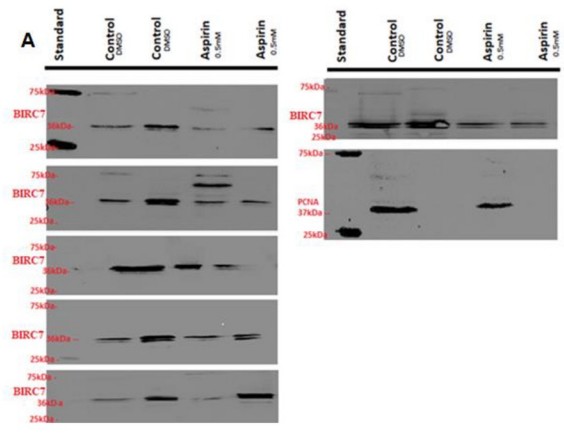

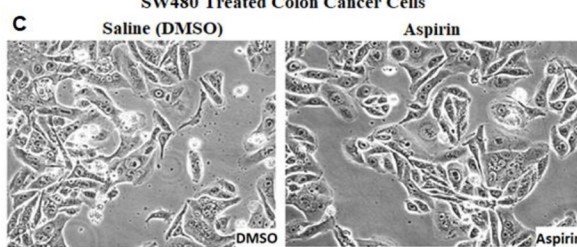

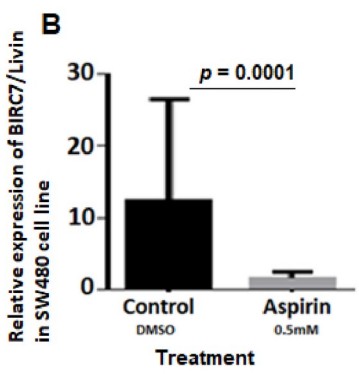

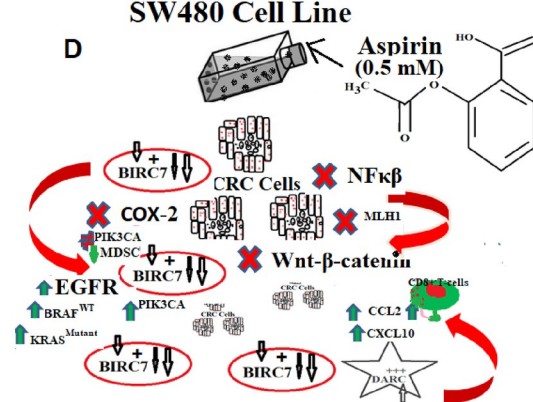

**Fig 2. Effect of aspirin treatment on BIRC7/Livin expression in DNA mismatch repair proficient, p53 mutant SW480 cell line *in vitro*. A** Is an Immunoblot showing effect of the 0.5 mM aspirin treatment on the SW480 compared to DMSO. PCNA was used as loading control. **B,** Quantitative analysis of BIRC7/Livin showing H score of the pattern of expression of the protein. **C)** SW480 cell line treated with 0.5 mM DMSO (vehicle control) and also with 0.5 mM aspirin for 24 hours (right). **D)** Describes the molecular pathway of aspirin metabolism, transport and target mechanises in colorectal cancer. This include inhibition of COX2, NF-kappaB and Wnt-B-Catenin signalling pathways for modulating BIRC7, DARC, PD-L1, Annexin V, PMS2, EGFR, BRAF, KRAS, PIK3CA, Bcl2, and p53 functions.

CRC (Fig 3C, 3E and 3F; $p = 0.0001$). The result also showed significant increase in expression of BIRC7/Livin in mucinous CRC post-FOLFOX treatment compared with pre-FOLFOX treatment (Fig 3C, 3D and 3I; $p = 0.0001$). There was significant suppression of BIRC7/Livin in CRC not otherwise specified post-FOLFOX treatment compared with pre-FOLFOX (Fig

**Table 1. Age and gender distribution of the colorectal cancer patients.**

| AGE (YEARS) | MALE | | FEMALE | | TOTAL | |
|---|---|---|---|---|---|---|
| | Number | % | Number | % | Number | % |
| **0–19** | 4 | 4.3 | 1 | 1.1 | 5 | 5.4 |
| **20–39** | 24 | 26.1 | 11 | 12.0 | 35 | 38.1 |
| **40–59** | 18 | 19.6 | 14 | 15.2 | 32 | 34.8 |
| **60–79** | 16 | 17.4 | 4 | 4.3 | 20 | 21.7 |
| Sum Total | **62** | **67.4** | **30** | **32.6** | **92** | **100** |

Mean age = 43.1±16.0.

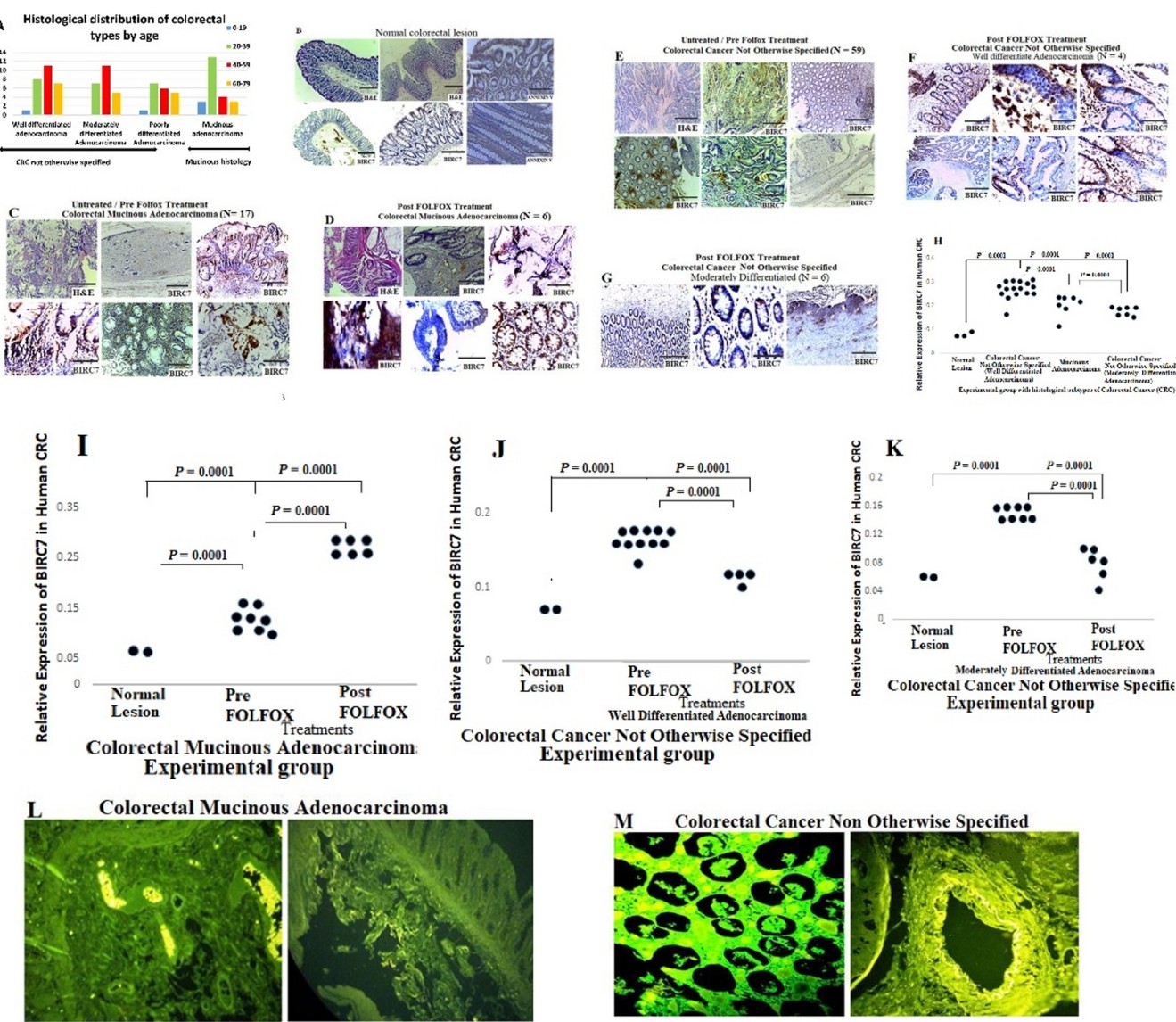

**Fig 3. The expression pattern and functional significance of BIRC7/Livin pre and post- FOLFOX treatment in human colorectal cancer (scale bar = 50 μm for H&E and scale bar = 100 μm for IHC). A)** A bar chart showing the age distribution of different CRC subtypes used in the study in Zaria Nigeria. **B)** Normal colorectal sections showing negative expression of BIRC7/Livin Scale bar, 20 um. **C)** Colorectal mucinous adenocarcinoma sections showing positive expression of BIRC7/Livin pre-FOLOFOX treatment, Scale bar, 20 um. **D)** Colorectal mucinous adenocarcinoma lesions showing increase expression of BIRC7/Livin post-FOLOFOX treatment. **E-G)** Colorectal cancer not otherwise specified (well and moderate differentiated) sections pre and post-FOLFOX chemotherapy showing increase expressions of BIRC7/Livin Scale bar, 20 um. **H-K)** BIRC7/Livin expression pattern typified in dot plot and showing pattern of the proteins expression using quantitative analysis technique. **L-M)** Representative IF images showing positive expression of BIRC7 in mucinous CRC histology and CRC not otherwise specified.

3E–3G, and 3J, 3K; *p* = 0.0001). The BIRC7/Livin expression was also evident by positive fluorescence stain in mucinous CRC and CRC not otherwise specified (Fig 3L and 3M).

Furthermore, we used a controlled *in vivo* model of NMU-induced CRC to study the prognostic significance of BIRC7 and the potential therapeutic importance of aspirin alone and aspirin plus FOLFOX in various combinations on the CRC in rat. Double contrast barium enema and histology analysis were employed to test the *in vivo* CRC cells growth, antitumor

activity and also the safety of the combination therapy models. Normal colorectal lesion show good uptake of the barium (Fig 4A) whereas diffuse and poor uptake of the barium was observed in CRC- bearing rats (Fig 4B). Result show free uptake of the barium in the normal colon lesion (Fig 4). Tumour involvement was evident by diffuse uptake of the barium at the anterior portion of the colon and little uptake at the mid to caudal portion (Fig 4B and 4C). Results also show disarrayed uptake of the barium post-aspirin plus FOLFOX treatment (Fig 4D). Grossed section of normal colon show areas occluded by faecal materials (Fig 4D). However, there is total absent of faecal materials in colonic lesion of the tumour-bearing rats pre-aspirin plus FOLFOX treatment (Fig 4F). The rats treated with aspirin plus FOLFOX in sequence show apparently normal colonic areas (Fig 4G), whereas concurrent treatment with aspirin plus FOLFOX show irregular distorted architecture of the colon and rectum with haemorrhagic areas (Fig 4H). Fig 5A–5M shows the treatment effect of aspirin with and without FOLFOX on apoptotic cell death in colorectal cancer in vivo in rat. Fig 5A–5F show various pattern of BIRC7/Livin suppression in aspirin with or without FOLFOX treated rats and also in untreated group. Fig 5G shows relative expression of the BIRC7 in rats typified in dot plots. Fig 5H is CRC-bearing rats treated with oxaliplatin showing obstructed colonic lumen devoid of faecal materials. Fig 5I show an enlarged abnormal Kidney (arrowed) from an oxaliplatin-treated rat observed during the post mortem. Fig 5J shows Histology analysis of the enlarged kidney showing features of renal cell carcinoma with sarcomatoid differentiation and positive

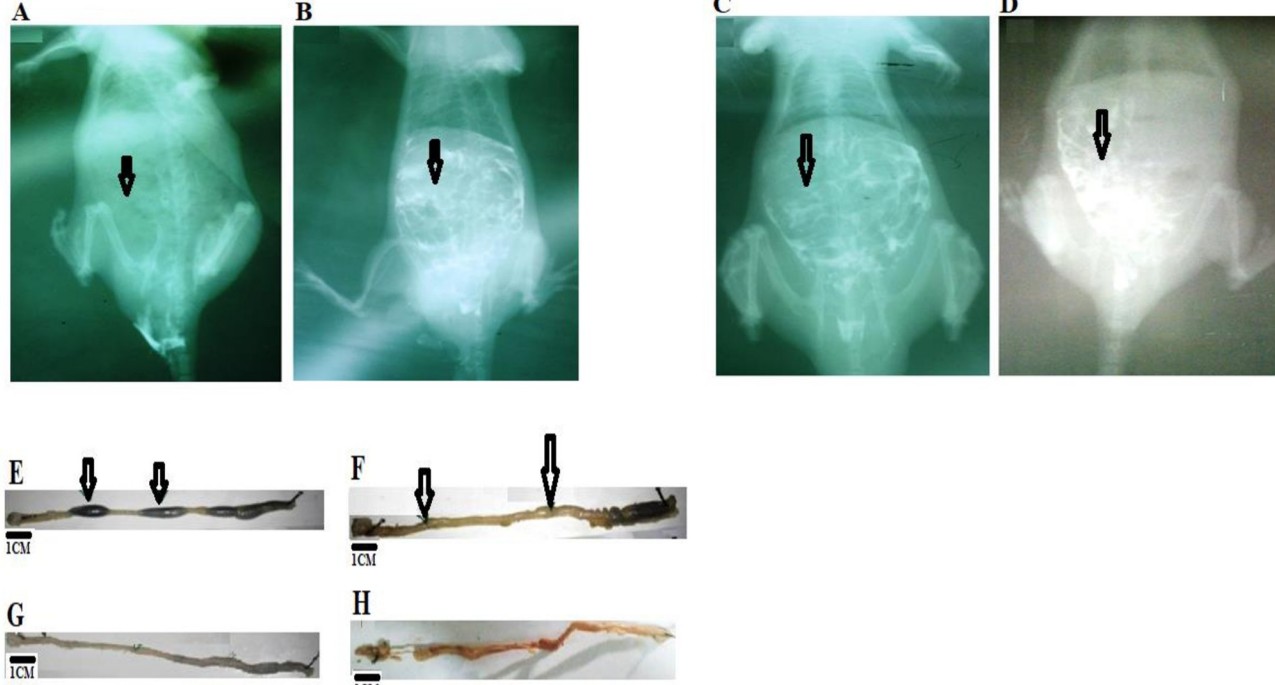

**Fig 4. Double contrast barium enema and grossed appearance of the colon and rectum post and pre-treatment in rats. A)** Apparently normal colon lesion showing free uptake of the barium. **B and C)** Tumour involvement was evident by diffuse uptake of the barium at the anterior portion of the colon and little uptake at the mid to caudal portion. This signifies tumour involvement. **D)** Disarrayed uptake of the barium post-aspirin plus FOLFOX treatment. **E,** Grossed section of apparently normal colorectal lesions with arrows pointing to areas occluded by faecal materials. **F)** Tumour-bearing colorectal lesions Pre aspirin plus FOLFOX treatment with arrows showing areas with tumour involvement and total absent of faecal material. **G)** Tumour-bearing colorectal lesion post-treatment with aspirin plus FOLFOX in sequence showing apparently normal areas. **H)** Tumour-bearing colorectal lesion post-concurrent treatment with aspirin plus FOLFOX showing irregular distorted architecture of the colon and rectum with haemorrhagic areas.

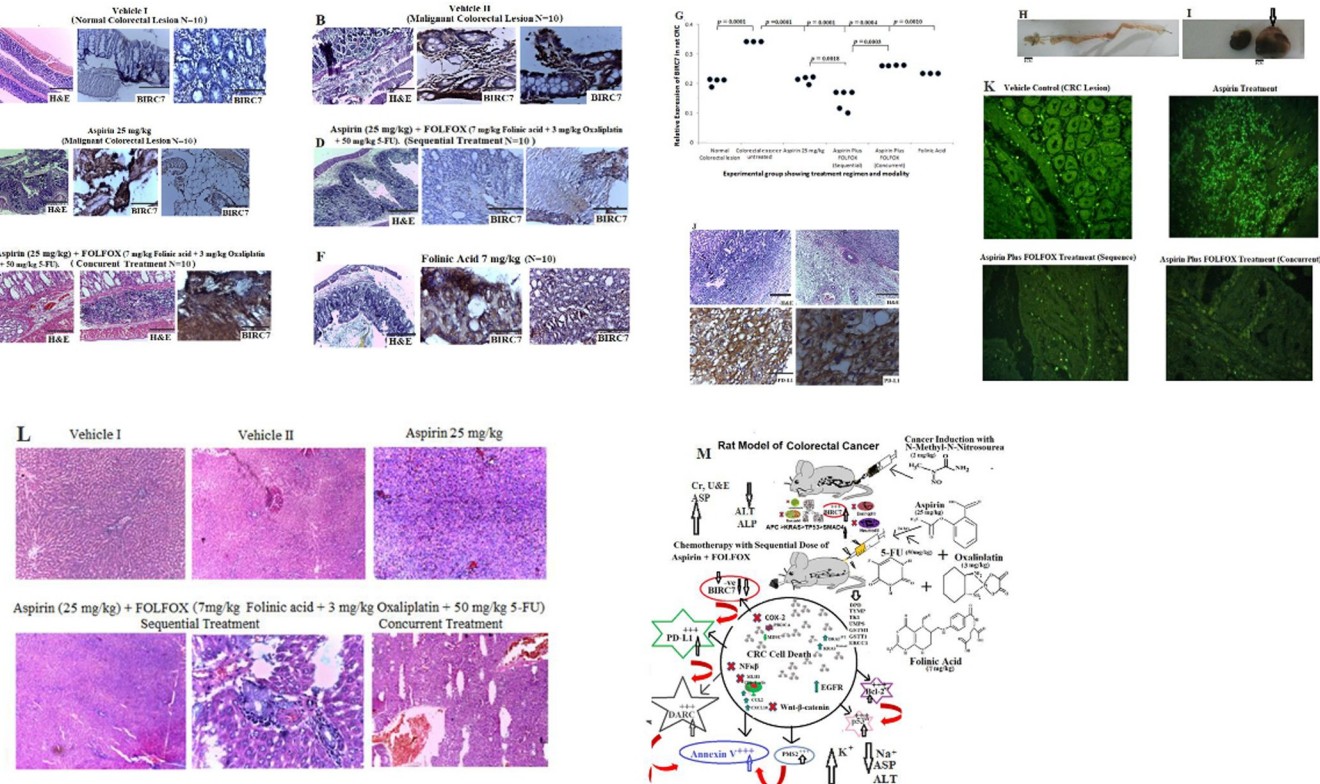

**Fig 5. The treatment effect of aspirin with or without FOLFOX on apoptotic cell death in colorectal cancer *in vivo* in rat (scale bar = 50 μm for H&E and scale bar = 100 μm for IHC). A—F** show various pattern of BIRC7/Livin expression in aspirin with or without FOLFOX treated rats and also in untreated group. **G** shows relative expression of the BIRC7 in rats typified in dot plot. **H** CRC-bearing rats treated with oxaliplatin showing obstructed colonic lumen devoid of faecal material. **I** show an enlarged abnormal Kidney (arrowed) from the oxaliplatin-treated rat observed during the post mortem. **J** Sections of enlarged kidney showing features of renal cell carcinoma with sarcomatoid differentiation with positive expression of PD-L1. **K** Sections showing different IF staining pattern upon treatment with aspirin only and aspirin plus FOLFOX in sequence and concurrent. **L** Sections of the liver showing preserved architecture that is devoid of haemorrhage in the group of rats treated with aspirin plus FOLFOX in sequence compared with concurrent treatment. **M** describes the possible pathways of the effect of sequential treatment of aspirin plus FOLFOX during tumour induction and chemotherapy. Several pathways are responsible for the aspirin and FOLFOX drug metabolism, transport and target mechanisms and they include but not limited to: COX2 inhibition, NFkB and Wnt-B-Catenin signalling pathways for aspirin; dihydropyrimidine dehydrogenase (DPD), thymidine phosphorylase (TYMP), thymidine kinase 1 (TK1) and uridine monophosphate synthetase (UMPS) for fluoropyrimidine component, 5-FU; and excision cross-complementing genes (ERCC) and glutathione S-transferases (GSTM) for the oxaliplatin (trans-1-diaminocyclohexane oxalateplatinum). These non-exhaustive have significant role in BIRC7, DARC, Annexin V, PD-L1, PMS2, EGFR, BRAF, KRAS, MDSC, PIK3CA, p53, BCL2 expression and functions. In NMU-induced colon carcinogenesis in rats, there is marked destruction of lymphocytes and granulocytes, significant reduction of red blood cells, platelet and total white blood cells. This was followed by reduction in alanine aminotransferase and alkaline phosphatase levels with increase in creatinine, electrolytes, urea, and aspartate aminotransferase levels. Sequential treatment of aspirin plus FOLFOX may confer a better anti-tumour immune response in CRC and this is justified by increase expression of PD-L, DARC and suppression of BIRC7.

for PD-L1 expression. Fig 5K shows sections with different IF staining pattern upon treatment with aspirin only and aspirin plus FOLFOX in sequence and concurrent. Fig 5L shows sections of the liver showing preserved architecture that is devoid of haemorrhage in the group of rats treated with aspirin plus FOLFOX in sequence compared with concurrent treatment. Fig 5M describes the entire pathophysiology of the effect of the sequential treatment of aspirin plus FOLFOX from tumour induction to chemotherapy. Overall, there were grossly clear multiple protrusions and absence of faecoliths in the colorectal lumen of the CRC-bearing rats (Fig 4F) compared with normal colon and rectum that was occupied by faecal materials (Fig 4D). This was accompanied by obstruction of colonic rugae and histologically detectable significant CRC cells with mucinous-like secretion compared with normal colon and rectum (Fig 5A and

5B; $p$ = 0.0001). There was a significant reduction in expression of BIRC7/Livin in the rat CRC cells post-treatment with 25 mg/kg aspirin ($p$ = 0.0001), sequential dose of aspirin plus FOL-FOX ($p$ = 0.0001), concurrent dose of aspirin plus FOLFOX (p = 0.0004), and Folinic acid alone ($p$ = 0.0010) compared with untreated CRC-bearing rats (Fig 5B–5F and 5K). The results also showed a significant reduction of BIRC7/Livin from the CRC-bearing rats treated with sequential dose of aspirin plus FOLFOX compared with aspirin treated rat ($p$ = 0.0018) and concurrent dose of aspirin plus FOLFOX-treated rats ($p$ = 0.0003). The colonic architecture of the CRC-bearing rat histologically and grossly appeared to be preserved after the sequential dose of aspirin plus FOLFOX treatment (see Fig 5D). In this sequentially treated group, the liver section in one of the rat appears preserved while another rat from this group shows necrotic features devoid of haemorrhage (Figs 4C and 5L). In addition, barium enema findings showed normal flow of the barium in the colon and rectum (Fig 4C). However, the results of the concurrent dose of aspirin plus FOLFOX treatment show grossly distorted colon and rectum with haemorrhagic features and absence of faecoliths compared with the other group (Fig 4E and 4H). This group shows haemorrhagic liver sections (Fig 5L) and the rat also appeared severely wasted and passing out watery and bloody stool with resulting poor survival outcome. The same features of severe necrosis were observed in the rats treated with Folinic acid alone (Fig 5B). One of the CRC-bearing rats treated with oxaliplatin alone showed an obstructed colonic lumen devoid of faecoliths (Fig 5H). An enlarged abnormal kidney was observed in this group (Fig 5I). Histological analysis of the enlarged kidney showed features of sarcomatoid renal cell carcinoma which was positive for PD-L1 protein expression (Fig 5J). There were variable florescence staining pattern on treatment with aspirin only and aspirin plus FOLFOX sequentially and concurrently (Fig 5K).

## Prognostic significance of p53, Bcl-2 and Annexin V in colorectal cancer cells pre and post-treatment with FOLFOX with or without aspirin

Dysregulations of p53 and Bcl-2 is associated with poor prognosis and chemoresistance in CRC [42], whereas Annexin V expression may stimulate the immunogenicity of cancer cells [26]. To determine whether p53 and Bcl-2 contribute to apoptotic activities, we probed the expression pattern of p53, Bcl-2 and Annexin V in human CRC pre and post-FOLFOX treatment and in rat-bearing CRC treated with aspirin alone and aspirin plus FOLFOLX in different combinations. Results from the human CRC cases showed positive expression of p53 and Bcl-2 in cases treated with neoadjuvant FOLFOX chemotherapy (Fig 6A–6D). There was no significant difference in the expression of p53 and Bcl-2 when comparing mucinous CRC with CRC NOS (Fig 6E and 6F, $p$ > 0.05). Analysis of Annexin V expression pattern showed positive expression of the protein in untreated and neoadjuvant FOLFOX-treated human CRC cases (Fig 6A–6D). This increase in expression of Annexin V may not be un-associated to tumour microenvironment and platelet reactions. Quantitative analysis of Annexin V demonstrated that there was a significant increase in protein expression among mucinous CRC pre-FOLFOX treatment compared with post-FOLFOX-treated cells ($p$ = 0.0001) and also post-FOLFOX-treated CRC NOS (well differentiated adenocarcinoma, $p$ = 0.0046). The results also showed a significant increase in the expression of Annexin V among post-FOLFOX-treated CRC NOS compared with post-FOLFOX-treated mucinous CRC ($p$ = 0.0019). These data suggest that there may be poor anti-tumour immune response which facilitates treatment resistance and evasion of apoptosis in human mucinous CRC treated with FOLFOX chemotherapy compared with CRC NOS.

Results from the *in vivo* study in rats, showed positive expression of p53 and Bcl-2 in sections from non CRC-bearing rats (Fig 7A), and also untreated CRC-bearing rats (Fig 7B),

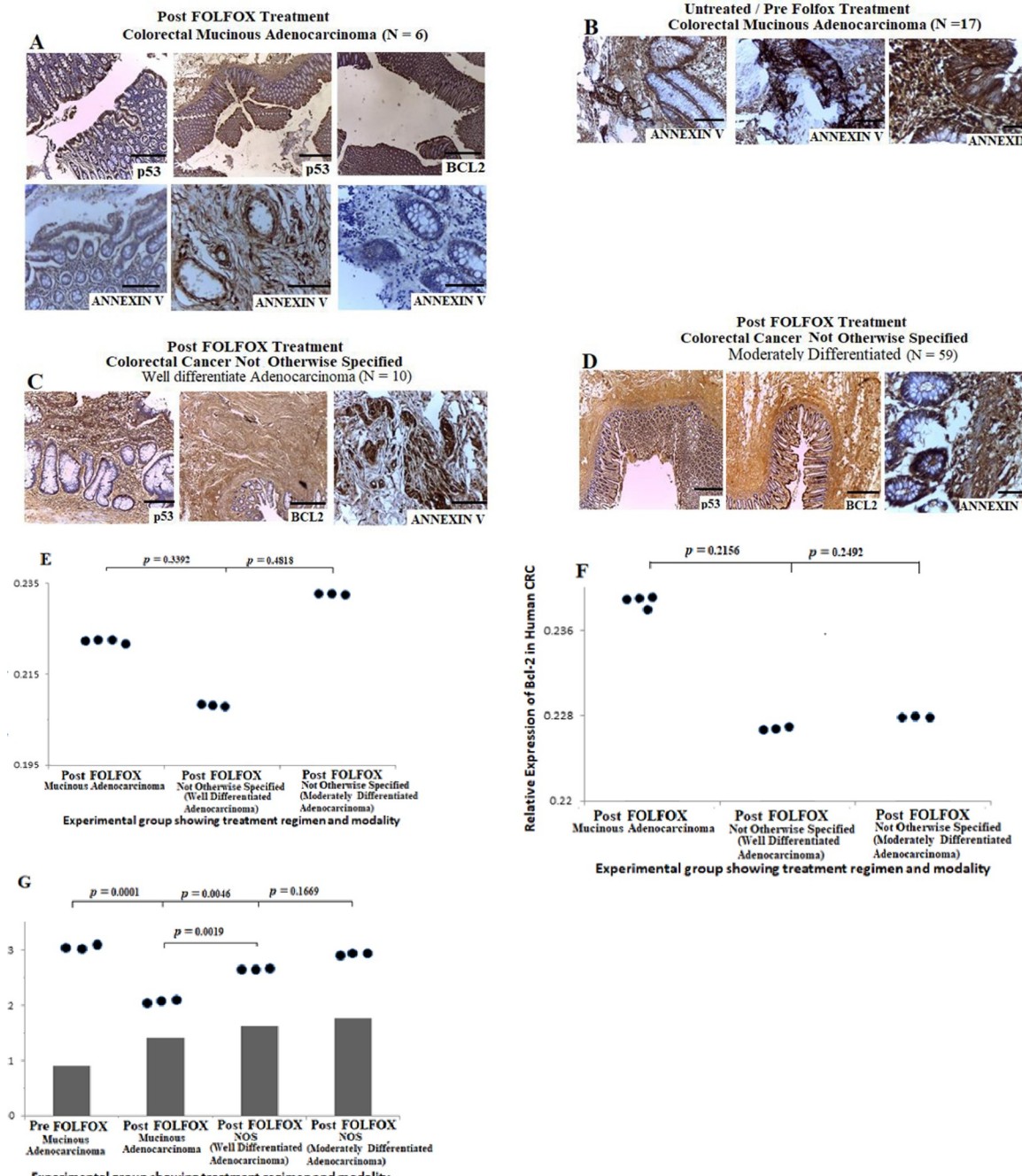

**Fig 6. Prognostic significance of p53, Bcl-2 and Annexin V in human colorectal cancer cells pre and post-treatment with FOLFOX n (scale bar = 50 μm for H&E and scale bar = 100 μm for IHC). A)** Colorectal mucinous adenocarcinoma sections showing positive expression of p53, Bcl-2 and Annexin V post-treatment with neoadjuvant FOLOFOX chemotherapy regimen, Scale bar, 20 um. **B)** Colorectal mucinous adenocarcinoma lesions showing increase in expression of Annexin V pre-FOLFOX treatment. **C—D)** Sections from colorectal cancer not otherwise specified (well differentiated and moderately differentiated types) post-FOLFOX chemotherapy showing strong positive expressions of p53, Bcl-2 and Annexin V. **E-G)** Quantitative analysis of p53, Bcl-2 and Annexin V showing the expression pattern typified in dot plot pre and post-treatment with FOLFOX.

treated with 25 mg/kg aspirin (Fig 7C), aspirin plus FOLFOX sequential (Fig 7D) and concurrent (Fig 7E) treatments, and folinic acid treated rats (Fig 7F). Quantitative analysis of p53

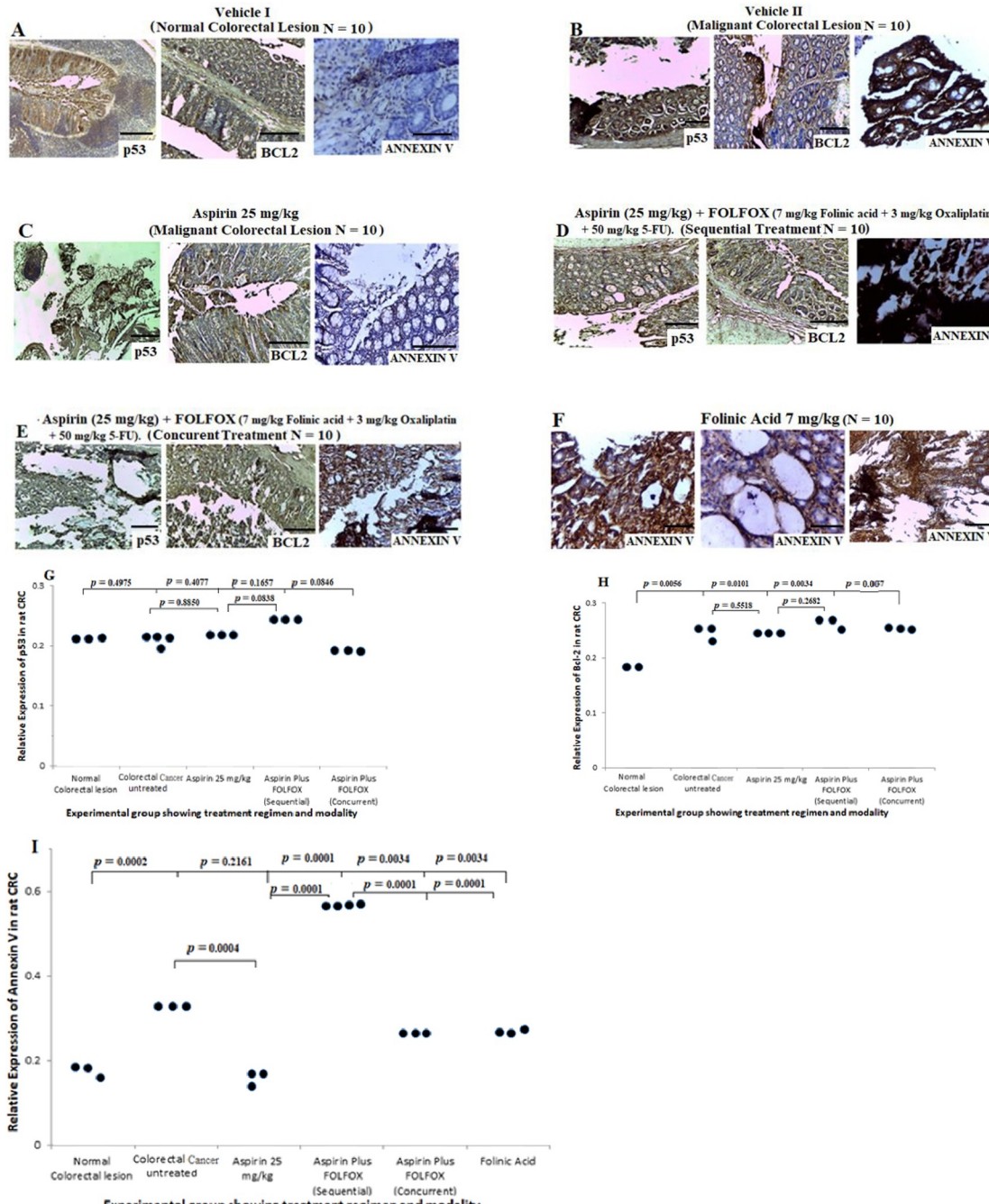

**Fig 7. Prognostic significance of p53, Bcl-2 and Annexin V in colorectal cancer cells post-treatment with FOLFOX with or without aspirin in rat (scale bar = 50 μm for H&E and scale bar = 100 μm for IHC). A)** Show positive expression of p53 and Bcl-2 and negative expression of Annexin V in normal colorectal sections. **B to F),** Show positive expression of p53, Bcl-2 and Annexin V in colorectal cancer sections pre and post-treatment with 25 mg/kg aspirin, aspirin plus FOLFOX in sequence, aspirin plus FOLFOX concurrent, and folinic acid treatment. **G to I)** Quantitative analysis of the p53, Bcl-2 and Annexin V from the rats showing expression pattern typified in dot plot post and pre-treatment with aspirin and aspirin plus FOLFOX in various combinations.

showed no significant difference in expression of the protein when comparing normal colon and rectum to CRC cells treated with different regimens. However, there was a significant

decrease in expression of Bcl-2 in normal colorectal section compared with sections from the untreated CRC rat ($p = 0.0056$), 25 mg/kg aspirin treated sections ($p = 0.0101$), aspirin plus FOLFOX sequential-treated sections ($p = 0.0034$) and aspirin plus FOLFOX concurrent-treated sections ($p = 0.007$). There is significant increase in expression of Annexin V in CRC-bearing rats untreated (Fig 7B, $p = 0.0002$), aspirin plus FOLFOX sequential (Fig 7D; $p = 0.0001$) and concurrent (Fig 7E; $p = 0.0034$) treatments, and folinic acid treated rats (Fig 7F; $p = 0.0034$) compared with sections from non CRC-bearing rats (Fig 7A). The quantitative analysis showed a significant increase in Annexin V expression in aspirin plus FOLFOX sequential-treated rats compared with 25 mg/kg aspirin treated rats ($p = 0.0001$) and aspirin plus FOLFOX concurrent-treated rats ($p = 0.0001$). There was no significance difference in Annexin V expression when comparing non CRC-bearing rats with aspirin treated rats (Fig 7A and 7C, $p = 0.2161$). Together, these data suggest that, in rat model of CRC, sequential treatment of aspirin plus FOLFOX may facilitate apoptotic and autophagic activities in CRC.

## Prognostic significance of PD-L1, DARC and DNA mismatch repair proteins MSH2 and PMS2 in human and rat models of CRC treated with FOLOFOX with or without aspirin

PD-L1 and DARC play important roles in tumour microenvironment by facilitating the escape of immunosurveillance and activation of tumor-specific immune response respectively, and these may influence cancer growth and metastasis [30, 32, 43]. The DNA MMR proteins MSH2 and PMS2 are key dysfunctional MMR proteins characterised by strong lymphocytic infiltrates [30, 44]. The DNA MMR proteins form part of the basis of microsatellite instability testing for identification of Lynch syndrome and immunotherapeutic target potential in patients with CRC [30, 44]. To determine the prognostic significance of PD-L1, DARC, MSH2 and PMS2 expression in CRC from West African Black patients in Zaria, and also from animal model, we conducted IHC staining on human CRC FFPE samples pre and post-FOLFOX chemotherapy and in rat-bearing CRC that were treated with aspirin and aspirin plus FOLFOX in various combinations. Results revealed negative expression of the PD-L1, DARC and PMS2 and positive expression of MSH2 in human mucinous CRC and CRC NOS pre and post-FOLFOX treatment (Fig 8A–8E). These data suggest that most of the Black African cases used in this study may harbour MSI-H CRC.

Results from the *in vivo* study in rats showed negative expression of PD-L1, DARC and PMS2 in sections from non CRC-bearing rats (Fig 9A). There was strong positive expression of PMS2 and negative expression of PD-L1 in the sections from the untreated CRC-bearing rats (Fig 9B). CRC sections from the rat treated with 25 mg/kg aspirin showed negative expression of PD-L1 and mild expression of PMS2 (Fig 9C). Sections from the rat-bearing CRCs treated with aspirin plus FOLFOX sequentially showed strong positive expression of PD-L1 and PMS2 with mild expression of DARC (Fig 9D). CRC sections treated with concurrent dose of aspirin plus FOLFOX showed mild expression of PD-L1 and DARC and strongly positive expression of PMS2 (Fig 9E). Sections treated with folinic acid showed focal expression of PD-L1, strong positive expression of DARC and mild expression of PMS2 (Fig 9F). Quantitative analysis of PD-L1 showed no significant difference in expression of PD-L1 when comparing normal colon and rectum with CRC- sections untreated ($p = 0.0909$), treated with 25 mg/kg aspirin ($p = 0.8214$), aspirin plus FOLFOX concurrent ($p = 0.9360$) and folinic acid ($p = 0.5912$) see Fig 9G. However, there was a significant increase in PD-L1 expression from CRC sections treated with sequential dose of aspirin plus FOLFOX compared with normal colon and rectum ($p = 0.0001$) and other treatment regimens (Fig 9G; $p = 0.0001$). There was also a significant increase in expression of DARC from sections treated with a sequential dose

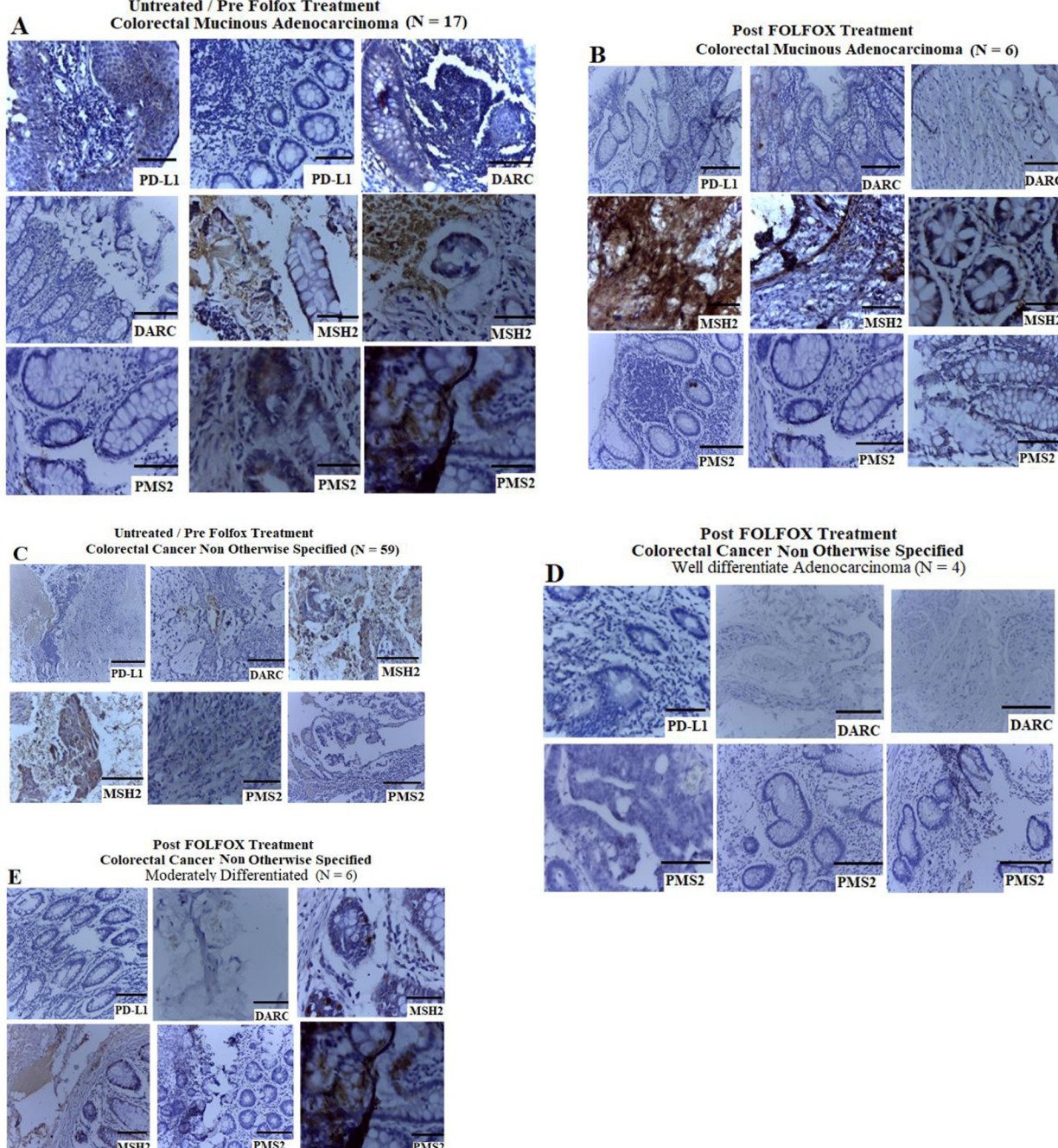

**Fig 8. Prognostic significance of PD-L1, DARC and DNA mismatch repair proteins MSH2 and PMS2 in human CRC treated with and without neoadjuvant FOLOFOX (scale bar = 50 μm for H&E and scale bar = 100 μm for IHC).** Colorectal adenocarcinoma sections showing negative expression of the PD-L1, DARC and PMS2 and positive expression of MSH2 in mucinous CRC pre and post-FOLFOX treatment (**A and B**), and pre and post-FOLFOX treated CRC NOS (**C—E**).

of aspirin plus FOLFOX compared with the normal sections ($p$ 0.0013) and also a concurrent dose of aspirin plus FOLFOX compared with the normal section (Fig 9H, $p$ = 0.0043). The quantitative analyses demonstrated a significant increase in PMS2 protein expression from the untreated CRC sections compared with sections treated with 25 mg/kg aspirin ($p$ = 0.0023)

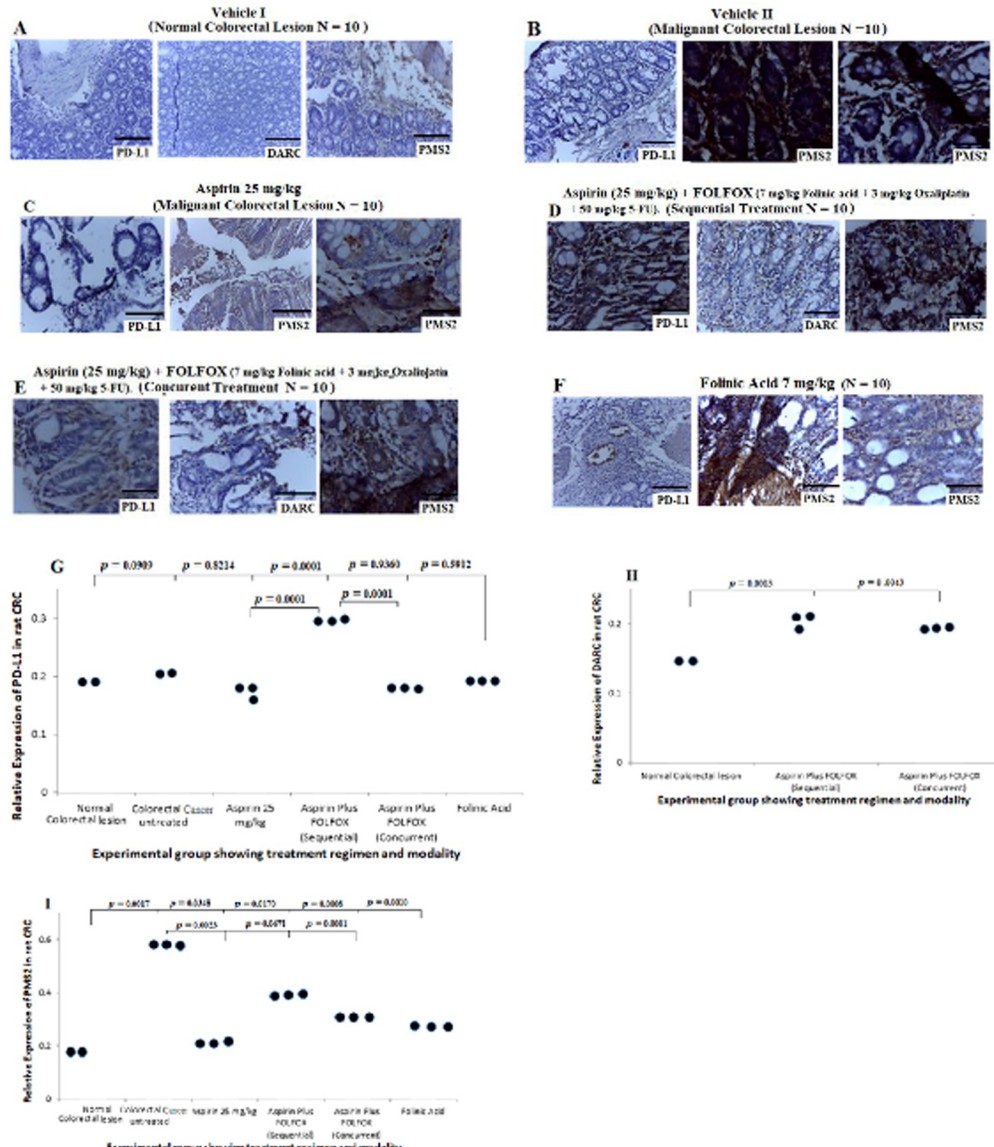

**Fig 9. Prognostic significance of PD-L1, DARC and DNA mismatch repair protein PMS2 in NMU-induced CRC in rat and treated with aspirin, and aspirin plus FOLOFOX in varous combinations (scale bar = 50 μm for H&E and scale bar = 100 μm for IHC). A)** Show negative expression of PD-L1, DARC and PMS2 in sections from non CRC-bearing rats. **B)** Show strong positive expression of PMS2 and negative expression of PD-L1 in the sections from the untreated CRC-bearing rats. C) CRC sections from the rat treated with 25 mg/kg aspirin showing negative expression of PD-L1 and mild expression of PMS2. D) Sections from aspirin plus FOLFOX sequential treated cells showing strongly positive expression of PD-L1 and PMS2 with mild expression of DARC. E) CRC sections treated with concurrent dose of aspirin plus FOLFOX showing mild expression of PD-L1 and DARC and strongly positive expression of PMS2. F) Sections treated with folinic acid showing focal expression of PD-L1, strong positive expression of DARC and mild expression of PMS2. G to I) Quantitative analysis of the PD-L1, DARC and PMS2 from the study showing expression pattern typified in dot plot post and pre-treatment with aspirin and aspirin plus FOLFOX in various combinations.

and aspirin plus FOLFOX concurrently ($p$ = 0.0081; Fig 9I). However, there was no significant difference in PMS2 expression from the untreated CRC sections compared with aspirin plus FOLFOX sequential treatment ($p$ = 0.0671). These data suggest that, sequential combination

treatment of aspirin plus FOLFOX in CRC may contribute to better anti-tumour immune response for effective activation of apoptosis and perhaps autophagy.

## Effect of colorectal carcinogenesis, FOLFOX and aspirin treatment on haematological and biochemical parameters in rats

Previous reports have implicated pre-treatment haematological parameters that include RBC, Hb, PLT, neutrophil count, and WBC as important prognostic factors for CRC management [45]. To determine the level and variation of haematological and biochemical parameters in prognostication of CRC, we studied these parameters in normal and CRC-bearing rats that were treated with different regimen of aspirin and aspirin plus FOLFOX in combination and in sequence. Results of the haematological parameters show significant decrease in red blood cells ($P = 0.0014$), haematocrit ($P = 0.0012$), platelet ($P = 0.0001$) and total white blood cells ($p = 0.0033$) from the CRC-bearing rats compared with CRC-free rats (Table 2). In comparing the differential WBC count between groups, we found a significant reduction in lymphocytes ($p = 0.0001$) and granulocytes ($p = 0.0001$) levels from the CRC–bearing rats compared with the CRC-free rats (Table 2). There were no significant difference in haematocrit and total white blood cell levels between untreated cancer-bearing rats and treated group (Table 3). The full blood count analysis using automation proved difficult in group of rats treated with neoadjuvant FOLFOX with or without aspirin because of prominent blood clot.

Results of the biochemical analysis showed significantly higher creatinine levels in the CRC-bearing rats compared with CRC-free rats (Table 4, $p = 0.0001$). There were significantly lower serum creatinine levels in the CRC-bearing rats treated with aspirin alone ($p = 0.001$), aspirin plus FOLFOX in sequence ($p = 0.003$), 5-FU alone ($p = 0.007$) and concurrent treatment of aspirin plus FOLFOX ($p = 0.0001$) compared with untreated CRC-bearing rats (Table 4). The results showed a significant increase in serum urea level from the CRC-bearing rats compared with CRC-free rats (Table 5, $p = 0.0075$). There were no significant difference in the serum urea level from the aspirin-treated group ($p = 0.9514$) and aspirin plus FOLFOX-treated rats sequentially ($p = 0.5488$) compared with CRC-free rats. However, the group of rats treated concurrently with aspirin plus FOLFOX showed significantly lower serum urea level compared with CRC-free rats ($p = 0.0316$). Treatment of the CRC-bearing rats with 5-FU alone showed significantly higher serum urea level compared with CRC-free rats ($p = 0.0017$).

There were significantly higher levels in serum electrolyte parameters, including potassium ($p = 0.0001$), sodium ($p = 0.0001$), chloride ($p = 0.0001$), bicarbonate ($p = 0.0001$) from the CRC-free rats compared with CRC-bearing rats (Table 5). There were no significant

**Table 2. Hematologic findings of rat pre and post colorectal cancer induction.**

| Blood components | Pre-cancer induction (Normal) | Post-cancer induction | Units | p-value[a] |
|---|---|---|---|---|
| Red blood cells | 7.31± .65 | 6.33± .96 | $10^{12}$/L | 0.0014 |
| Haemoglobin | 14.28± 1.14 | 12.30± 2.52 | g/dL | 0.1302 |
| Haematocrit | 40.41± 3.46 | 37.62± 7.08 | % | 0.0012 |
| Platelet | 513.04± 98.15 | 370.36± 129.73 | $10^9$/L | 0.0001 |
| Total white blood cells | 11.06± 3.57 | 3.01± .61 | $10^9$/L | 0.0033 |
| Differentials | | | | |
| Lymphocytes | 8.83± 2.92 | 2.38± .56 | $10^9$/L | 0.0001 |
| Granulocytes | 1.10± .63 | 0.25±.12 | $10^9$/L | 0.0001 |

Values are presented as mean±SD.

[a]The result is significant when $p < 0.05$.

**Table 3. Post-treatment values of haematocrit and total white blood cells parameters of colorectal cancer-bearing rats.**

| Treatment | Haematocrit (%) | Total white blood cells ($10^9$/L) |
|---|---|---|
| Cancer induced | 37.62± 7.08 | 3.01± .61 |
| Aspirin alone | 41.00± 7.55 | 6.9± 4.31 |
| *p value* | 0.6019 | 0.1966 |
| Cancer induced | 37.62± 7.08 | 3.01± .61 |
| Aspirin plus FOLFOX (Sequentially) | 38.33± 3.51 | 12.83± 10.32 |
| *p value* | 0.8839 | 0.1753 |
| Cancer induced | 37.62± 7.08 | 3.01± .61 |
| Aspirin plus FOLFOX(Concurrently) | 38.67± 10.3 | 3.83± 2.53 |
| *p-value* | 0.8913 | 0.6143 |
| Cancer induced | 37.62± 7.08 | 3.01± .61 |
| 5-FU alone | 45.33± 2.0 | 22.36± 22.69 |
| *p-value* | 0.1437 | 0.2138 |

Values are presented as Mean±SD.

differences in serum potassium level from aspirin-treated CRC-bearing rats compared with untreated rats ($p = 1.0000$). The results also show significantly lower serum sodium ($p = 0.0165$), chloride ($p = 0.0053$) and bicarbonate ($p = 0.0001$) levels from the aspirin-treated CRC-bearing rats compared with CRC-free rats. The group of rats treated with aspirin plus FOLFOX sequentially showed significantly higher serum potassium ($p = 0.0332$), and lower sodium ($p = 0.0001$) and chloride ($p = 0.0001$) levels compared with the CRC-free rats. There were no significant differences in serum bicarbonate levels between the CRC-bearing rats treated sequentially with aspirin plus FOLFOX compared with the CRC-free rats ($p = 0.8013$). Results from the CRC-bearing rats treated concurrently with aspirin plus FOLFOX showed significantly low serum sodium ($p = 0.0001$) and chloride ($p = 0.0001$) levels and a higher bicarbonate level ($p = 0.0001$) compared with the CRC-free rats. There were no significant difference in potassium level ($p = 0.1940$) from this group of rats compared with the CRC free group (Table 5). In addition, there was a significant difference in serum potassium

**Table 4. Creatinine values of colorectal cancer-bearing rats treated with aspirin with or without neo-adjuvant FOLFOX.**

| Treatments | Creatinine values (umol/L) | *p*-value[a] |
|---|---|---|
| Normal | 33.3±26.5 | 0.0001 |
| Cancer induced | 65.00±8.48 | |
| Cancer induced* | 65.00±8.48 | |
| Aspirin alone | 4.75± 1.50 | 0.001 |
| Cancer induced | 65.00±8.48 | |
| Aspirin plus FOLFOX (Sequentially) | 4.50±2.12 | 0.003 |
| Cancer induced | 65.00±8.48 | |
| Aspirin plus FOLFOX (Concurrently) | 3.40± .89 | 0.0001 |
| Cancer induced | 65.00±8.48 | |
| 5-FU alone | 11.00±5.65 | 0.007 |

Values are presented as mean±SD.
[a]The result is significant when $p < 0.05$.

**Table 5. Biochemical findings of various group of rats induced with colorectal cancer and treated with Aspirin plus FOLFOX chemotherapy in various combinations.**

| Treatments | Serum urea and electrolyte | | | | | Liver function test | | |
|---|---|---|---|---|---|---|---|---|
| | Urea (mmol/L) | Potassium (mmol/L) | Sodium (mmol/L) | Chloride (mmol/L) | HCO$_3^{-*}$ (mmol/L) | AST** (UI/L) | ALT**** (UI/L) | ALP**** (UI/L) |
| **Normal (CRC free rats)** | **3.4±1.6** | **5.7±.9** | **136.3±1.5** | **101.6±2.8** | **18.0±3.6** | **146.0±26.9** | **62.0±14.0** | **845.0±208.9** |
| **CRC- induced** | 5.3±.3 | 4.9±.3 | 132.5±.7 | 94.5±.7 | 9.5±6.4 | 190.0±91.9 | 33.0±12.7 | 648.5±55.8 |
| *p* value[a] | 0.0075 | 0.0001 | 0.0001 | 0.0001 | 0.0001 | 0.0260 | 0.0001 | 0.0001 |
| **Aspirin alone** | 3.3±.8 | 5.7±1.2 | 133.8±4.8 | 98.5±4.5 | 31.0±11.8 | 145.5±63.1 | 61.8±22.9 | 725.5±575.1 |
| *p*-value[a] | 0.9514 | 1.0000 | 0.0165 | 0.0053 | 0.0001 | 0.9711 | 0.9704 | 0.3337 |
| **Aspirin + FOLFOX (Sequentially)** | 3.1±1.9 | 7.2±3.3 | 122.0±15.5 | 87.0±11.3 | 18.5±9.2 | 107.5± 23.3 | 38.0±8.5 | 755.5±400.9 |
| *p*-value[a] | 0.5488 | 0.0332 | 0.0001 | 0.0001 | 0.8013 | 0.0001 | 0.0001 | 0.3272 |
| **Aspirin + FOLFOX (Concurrently)** | 2.3±1.9 | 6.5±2.9 | 127.0±7.9 | 89.0±8.2 | 23.4±4.6 | 148.4±41.4 | 40.4±12.5 | 329.0±61.9 |
| *p*-value[a] | 0.0316 | 0.1940 | 0.0001 | 0.0001 | 0.0001 | 0.8090 | 0.0001 | 0.0001 |
| **5-FU alone** | 6.7±4.7 | 7.9±3.5 | 121.5±10.6 | 85.0±7.0 | 23.5±3.5 | 273.5 ±174.6 | 77.0±35.4 | 296.0±18.4 |
| *p* value | 0.0017 | 0.0038 | 0.0001 | 0.0001 | 0.0001 | 0.0007 | 0.0546 | 0.0001 |

Values are presented as mean±SD. Each parameter of the treatment type was compared with the cancer induced group.

[a]The result is significant when *p* <0.05.

*HCO$_3$- (Bicarbonate).

** AST (Aspartate aminotransferase).

*** ALT (Alanine aminotransferase).

****ALP (Alkaline phosphatase).

(*p* = 0.0038) and bicarbonate (*p* = 0.0001) levels as well as serum sodium (*p* = 0.0001) and chloride (*p* = 0.0001) levels from the group of CRC-bearing rats treated with 5-FU alone compared with the CRC-free rats (Table 5).

Results of the liver function test parameters showed significantly higher aspartate aminotransferase (*p* = 0.0260) and lower alanine aminotransferase (*p* = 0.0001) and alkaline phosphatase (*p* = 0.0001) levels in CRC-bearing rats compared with the CRC-free rats (Table 5). The results showed no significant difference in aspartate aminotransferase (*p* = 0.9711), alanine aminotransferase (*p* = 0.9704) and alkaline phosphatase level (*p* = 0.3337) in the group of CRC-bearing rats treated with aspirin compared with the CRC-free rats. There was significantly lower aspartate aminotransferase (*p* = 0.0001) and alanine aminotransferase (*p* = 0.0001) levels in the group of CRC-bearing rats treated in sequence with aspirin plus FOLFOX compared with the CRC-free rats. However, the group of CRC-bearing rats treated concurrently with aspirin plus FOLFOX showed significantly lower alanine aminotransferase (*p* = 0.0001) and alkaline phosphatase (*p* = 0.0001) levels, compared with the CRC-free rats. The results also showed a significant increase in aspartate aminotransferase (*p* = 0.0007) and lower alkaline phosphatase (*p* = 0.0001) levels from the CRC-bearing rats treated with 5-FU alone compared with the CRC-free rats (Table 5). Together, these data suggest that, destruction of immune cells may pave way for CRC carcinogenesis.

## Discussion

The field of cancer genetics and cancer biology is providing remarkable advances in understanding the consequence of disruption of apoptosis in promoting treatment resistance and metastasis in CRC [9, 10]. Cancer chemotherapy regimens such as FOLFOX have been shown

to significantly increased treatment efficacy and improved survival in CRC however, this treatment modality may be accompanied by complications such as myelotoxicity, neurotoxicity, non-alcoholic fatty liver disease and sinusoid obstruction syndrome [46]. Aspirin has been shown to be effective in preventing recurrence, decreasing risk of metastasis post curative therapy and primary prevention of CRC [47]. Complications associated with aspirin use -such as occult gastrointestinal bleeding, epistaxis extra and intra-cranial bleeding which, are very rare may resolve without much ramification [48]. There is limited research that explores the mechanism through which aspirin alone and in combination with other chemotherapeutics exert anticancer effect especially in Black Africans with CRC. Thus, this strengthen the rational for testing Aspirin plus FOLFOX towards identifying biomarkers and clinical characteristics which may predict the benefit of Aspirin use for targeted invention in CRC management. Our study uncovers that mucinous CRC may serve as comfortable niche for inhibitor of apoptosis proteins such as BIRC7/Livin to strive and facilitate treatment resistance, local recurrence and metastasis especially after neoadjuvant chemotherapy with FOLFOX. Despite advances in CRC management, prognosis has been poor in some patients, with reports indicating that 20% and 30% of patients present with metastasis at diagnosis and after curative intent respectively [49, 50]. Previous reports have shown that mucinous CRC, which is more common in White than Asian individuals, presents with increased lymph node involvement, metastatic features and advanced stage at diagnosis and decrease overall survival compared with NOS CRC [51, 52]. Our findings from rural Africans with CRC indicate low expression of PD-L1, DARC and PMS2 and positive expression of MSH2 in mucinous histology CRC and CRC NOS pre and post-FOLFOX chemotherapy. Microsatellite instability-high CRC (CRC with negative expression of either one or two of MMR proteins—MSH2, PMS2, MLH1, and MSH6) has been shown to be highly immunogenic and harbour mutation-associated genes [52]. Mucinous CRC has also been shown to present more often with microsatellite instability, BRAF and PIK3CA mutations than CRC not otherwise specified [52]. A recent finding also indicated that metastasis in CRC may occur much earlier before tumour diagnosis as a result early driver genes mutations such as APC, KRAS, TP53 and SMAD4 mutations [53]. Our results further show positive expression of anti-apoptotic Bcl-2 and p53, followed by significant increase in Annexin V protein expression in mucinous CRC pre-FOLFOX treatment compared with post-treatments and also CRC NOS ($p < 0.05$). The p53 overexpression has been associated with poor prognosis and resistance to chemotherapy in CRC [42]. This suggests that there may be poorer chemotherapeutic effect on mucinous CRC than CRC NOS and this can result in a weak anti-tumour immune response and treatment resistance. One possible explanation for these observations may not be unrelated to the fact that mucinous CRC is more commonly localized in the proximal colon and diagnosed at advanced stage than CRC NOS. In addition, abundant mucous layer from MUC-2 protein expression which hinders the activity of antitumour immune response may also contribute to treatment resistance in mucinous CRC [54, 55]. Mucinous CRC has also been reported to respond poorly to 5-FU based treatment as a result of overexpression of DNA topoisomerase 1, microsatellite instability and abundant extracellular mucin which create a physical barrier that reduce chemotherapeutic efficacy [56, 57]. However, a different report has interestingly documented improved prognosis in patients with increased expression of MUC-2 protein in mucinous CRC than CRC NOS [58]. In essence, sequential treatment of patients with aspirin may chemosensitizes to FOLFOX regimen in management.

The *in vitro* study indicates a significant suppressive effect of 0.5 mM aspirin on BIRC7/Livin and reduction in CRC cell growth in the SW480 cell line. This finding is consistent with previous reports in the HT29 cell line model [59]. Conversely, small interfering RNA (siRNA) transfection has been reported to downregulate BIRC7/Livin expression towards apoptotic induction in CRC *in vitro* and *in vivo* [60]. Aspirin reduces CRC growth via different

mechanisms such as, inhibition of cyclooxygenase-2, dysregulation of NFκβ and Wnt-β-cate-nin signalling pathways, activation of EGFR pathway, host immune response modulation and up-regulation of tumour suppression genes [61–63]. Our findings using the TCGA showed a higher expression of BIRC7/Livin in Whites and Asians compared with African-Americans. However, CRC cases among racial/ethnic minorities in TCGA were limited such that there may be insufficient power to discern detectable differences in this dataset. There is variation in survival probability when comparing high and low expression pattern of BIRC7/Livin among White, African American and Asian individuals with CRC from the TCGA. Post diagnostic CRC survival rates have been reported to be worse among young black patients than whites [64]. Moreover, recent findings that utilized the Surveillance, Epidemiology, and End Results (SEER)- Medicare database show 16–21% increase CRC mortality in African Americans compared with Whites, and 13% and 20% lower mortality in Asians and Hispanics respectively [65]. However, these results are for an older population and given the median age of CRC in the Nigerian cohort being much younger, this may also be a limitation in extrapolating these findings to Black African cases in Nigeria and supporting the need for future studies to understand the role of these genes as prognostic markers among Nigerians.

Furthermore, we demonstrated that sequential treatment of aspirin plus FOLFOX confers a better anti-tumour immune response in CRC. This leads to effective suppression of BIRC7 without causing clear damage to the luminal colonic area. The treatment also did not cause haemorrhage in the liver of rats compared with concurrent treatment model which showed marked tumour metastatic features and organ damage. BIRC7/Livin has been reported to facilitate tumour cell metastasis in CRC by evasion of apoptosis and autophagy [40, 41]. Our result showed a significant increase of Annexin V protein expression in rats treated sequentially with aspirin plus FOLOX compared with other treatment modalities ($p < 0.05$). In addition, there was a significant increase in PD-L1 and DARC expression from CRC sections treated with sequential dose of aspirin plus FOLFOX compared other treatment combinations. Moreover, overexpression of PD-L1 and the chemokine receptor, DARC, have been reported to correlates with better prognosis among cancer cases, including CRC and particularly among MMR-proficient CRC patients [32, 33, 66]. The cytotoxic CD8+ T-cells have been reported to be increased by the effect of aspirin via the alteration of chemokines which also contribute to decrease in myeloid-derived suppressor cells towards modulation of apoptosis [67]. These findings suggest that treatment of CRC with sequential dose of aspirin plus FOLFOX confer better anti-tumour immune response and may effectively induce apoptosis and autophagy.

Lastly, we demonstrated that in NMU-induced colon carcinogenesis in rats, there is marked destruction of lymphocytes and granulocytes, significant reduction of red blood cells, platelet and total white blood cells. This was followed by reduction in alanine aminotransferase and alkaline phosphatase levels with increase in creatinine, electrolytes, urea, and aspartate aminotransferase levels compared with normal rats (p < 0.05). However, sequential treatment of the CRC-bearing rats indicated significantly higher serum potassium, and low sodium chloride, aspartate aminotransferase and alanine aminotransferase levels compared to normal rats. Our report suggests that, destruction of immune cells may pave way for CRC carcinogenesis. Analysis of the blood samples post aspirin plus FOLFOX treatment proved difficult with automation. This implies that this treatment modality may be associated with significant hematologic and biochemical toxicities and patients on this combination of treatment must be closely monitored and managed for these derangements when and if they arise.

In summary, our study provides new insights into the prognostic role of BIRC7/Livin expression in evasion of apoptosis and facilitation of treatment resistance, and metastasis in mucinous CRC -specifically post-neoadjuvant FOLFOX chemotherapy. These poor prognostic features in CRC are further compounded by suppression of DARC, PD-L1, PMS2 and

expression of MSH2 in rural African patients with CRC. The mechanistic rationale of aspirin plus FOLFOX sequential combination therapy in suppressing BIRC7/Livin and reactivating DARC and PD-L1 expression underscores the potential of aspirin combination therapy in future CRC management. Our findings support a randomized clinical trial of aspirin in combination with FOLFOX sequentially in patients with CRC. However, because of the significant hematologic and biochemical toxicities associated with aspirin plus FOLFOX chemotherapy, special precaution needed to be taken when these toxicities are observed in humans and should be appropriately managed. The use of granulocyte colony stimulating factor [G-CSF] may be considered during chemotherapy as this will go a long way in protecting patients from developing severe grade toxicity. Future investigations using advance genomic and transcriptomic analysis of the role of aspirin as single agent and in combination with FOLFOX in human CRC is needed to fully understand patient response to this therapeutic approach. These investigations should take into consideration post aspirin plus FOLFOX drug-drug interactions, immunomodulatory responses and epigenetic factors among others.

## Supporting information

**S1 Appendix. Tabular presentation of treatment composition in the rat-bearing CRC.**
(DOCX)

**S1 Raw images.**
(PDF)

## Acknowledgments

The Authors wish to thank Salihu Abdullahi (late) and Ahmed Aliyu from Faculty of Pharmaceutical Sciences Ahmadu Bello University Zaria, Nigeria for role in carcinogenesis study; Dr I. D Nicholl of the University of Wolverhampton, England, United Kingdom for mentoring and guidance in the in vitro study. We are grateful to Dr Andreana Holowatyj for her helpful comments on the manuscript. We are grateful to Hannatu Aminu and Rukayya Abdullahi from Department of Hematology and Blood Transfusion, Ahmadu Bello University Teaching Hospital Zaria for haematology studies; Jigo Dangude Yaro, James O Enamari, Abdullahi A Mairiga, Lucky I Aghemunu, Fatima Ladan and Umar Lawal from Department of Pathology, Ahmadu Bello University Zaria, for histology and immunohistochemistry studies; Dr Lawal Maruf from Department of Veterinary Surgery and Radiology, Ahmadu Bello University Zaria for Radiology and Barium Enema study; and Jonathan Aluwong from Department of Chemical Pathology, Ahmadu Bello University Teaching Hospital Zaria, Nigeria for biochemical analysis.

These findings were presented in part at the following conferences: American Association for Cancer Research (AACR) Annual Meeting on March 29th–April 5th 2019 in Atlanta Georgia USA; AACR Annual Meeting April 1–5, 2017 in Washington DC; AACR Special Conference on Precision Medicine Series: Targeting the Vulnerabilities of Cancer on May 16–19, 2016 in Miami, Florida and; AACR New Horizons in Cancer Research Conference: Bringing Cancer Discoveries to Patients held from 12 to 15 November 2015 in Shanghai, China.

## Author Contributions

**Conceptualization:** Mohammed Faruk, Sani Ibrahim.

**Data curation:** Mohammed Faruk, Halimatu Sadiya Musa.

**Formal analysis:** Mohammed Faruk, Khalid Zahir Shah, Kasimu Umar Adoke.

**Funding acquisition:** Mohammed Faruk.

**Investigation:** Mohammed Faruk, Surajo Mohammed Aminu, Ahmed Adamu, Adamu Abdullahi, Aishatu Maude Suleiman, John Idoko, Kasimu Umar Adoke.

**Methodology:** Mohammed Faruk, Sani Ibrahim, Surajo Mohammed Aminu, Ahmed Adamu, Adamu Abdullahi, Abdullahi Mohammed, Yawale Iliyasu, John Idoko, Halimatu Sadiya Musa, Sani Abubakar, Kasimu Umar Adoke, Cheh Agustin Awasum.

**Project administration:** Mohammed Faruk, Adamu Abdullahi, Cheh Agustin Awasum.

**Resources:** Adamu Abdullahi, Abdulmumini Hassan Rafindadi, Abdullahi Mohammed, Yawale Iliyasu, John Idoko, Rakiya Saidu, Abdullahi Jibril Randawa, Atara Ntekim, Khalid Zahir Shah, Muhammad Manko.

**Supervision:** Mohammed Faruk, Sani Ibrahim, Surajo Mohammed Aminu, Ahmed Adamu, Atara Ntekim, Cheh Agustin Awasum.

**Validation:** Mohammed Faruk, Aishatu Maude Suleiman.

**Writing – original draft:** Mohammed Faruk, Ahmed Adamu, Adamu Abdullahi, Aishatu Maude Suleiman.

**Writing – review & editing:** Mohammed Faruk, Sani Ibrahim, Surajo Mohammed Aminu, Ahmed Adamu, Adamu Abdullahi, Aishatu Maude Suleiman, Abdulmumini Hassan Rafindadi, Abdullahi Mohammed, Yawale Iliyasu, John Idoko, Rakiya Saidu, Abdullahi Jibril Randawa, Halimatu Sadiya Musa, Atara Ntekim, Khalid Zahir Shah, Sani Abubakar, Kasimu Umar Adoke, Muhammad Manko.

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
