## [Decision Letter · Decision Letter 0]

15 Sep 2020

PONE-D-20-18593

Prognostic Significance of BIRC7/Livin, Bcl-2, p53, Annexin V, PD-L1, DARC, MSH2 and PMS2 in Colorectal Cancer Treated with FOLFOX Chemotherapy with or without Aspirin

PLOS ONE

Dear Dr. Faruk,

Thank you for submitting your manuscript to PLOS ONE. After careful consideration, we feel that it has merit but does not fully meet PLOS ONE’s publication criteria as it currently stands. Therefore, we invite you to submit a revised version of the manuscript that addresses the points raised during the review process.

The manuscript should be edited for scientific clarity, including providing a clear rationale.

Additional details have to be provided on the aspirin trial.

We look forward to receiving your revised manuscript.

Kind regards,

Irina V. Lebedeva, Ph.D.

Academic Editor

PLOS ONE

Journal Requirements:

2. In the ethics statement in the manuscript and in the online submission form, please provide additional information about the patient records used in your retrospective study, including: a) the date range (month and year) during which patients' medical records were accessed and b) the date range (month and year) during which patients whose medical records were selected for this study sought treatment.

3.At this time, we request that you  please report additional details in your Methods section regarding animal care, as per our editorial guidelines:

(1) Please state whether the provided ethics committee contains animal welfare experts or whether an animal ethics or IACUC committee reviewed and approved the study. Please provide the full name of the committee that reviewed and approved the study  

(2) Please state the source of the rats used in the study

(3) Please describe any steps taken to minimize animal suffering and distress, such as by administering anaesthesia, during the blood collection via venepuncture of the abdominal aorta in the rats.   

(4) Please describe the care received by the animals, including the frequency of monitoring and the criteria used to assess animal health and well-being during the course of the experiment

Thank you for your attention to these requests.

4. For studies involving humans categorized by race/ethnicity, authors should update outmoded terms and potentially stigmatizing labels to more current, acceptable terminology. For instance, “Caucasian” should be changed to “white” or “of [Western] European descent” (as appropriate).

5. In the Methods section, please provide the source, product number and any lot numbers of the N-Methyl-N-Nitrosourea (NMU) used in the animal experiments for your study.

6. In the Methods section, please provide the source, product number and any lot numbers of the primary antibodies used in the histology analysis  for your study.

7. To comply with PLOS ONE submission guidelines, in your Methods section, please provide additional information regarding your statistical analyses. For more information on PLOS ONE's expectations for statistical reporting, please see https://journals.plos.org/plosone/s/submission-guidelines.#loc-statistical-reporting.

8. At this time, we ask that you please provide scale bars on the microscopy images presented in the Figures and refer to the scale bar in the corresponding Figure legend.

9. Please include your tables as part of your main manuscript and remove the individual files. Please note that supplementary tables (should remain/ be uploaded) as separate "supporting information" files

Reviewers' comments:

Reviewer's Responses to Questions

**Comments to the Author**

1. Is the manuscript technically sound, and do the data support the conclusions?

Reviewer #1: Yes

Reviewer #2: Partly

2. Has the statistical analysis been performed appropriately and rigorously? 

Reviewer #1: Yes

Reviewer #2: I Don't Know

3. Have the authors made all data underlying the findings in their manuscript fully available?

Reviewer #1: Yes

Reviewer #2: Yes

4. Is the manuscript presented in an intelligible fashion and written in standard English?

Reviewer #1: Yes

Reviewer #2: No

5. Review Comments to the Author

Reviewer #1: A mountain of work presented here impressively presented with interpretations provided

It would be helpful to have more details of the 92 colorectal cancers in the aspirin trial - age, tumour location in the bowel, stage, microsatellite stability in view of the ASPREE study, were there any age related effects on the biomarker outcomes?

Was there a differential effect of aspirin seen in the MSS vs MSI-H patients or models?

Reviewer #2: PLOS ONE

Prognostic Significance of BIRC7/Livin, Bcl-2, p53, Annexin V, PD-L1, DARC, MSH2 and PMS2 in Colorectal Cancer Treated with FOLFOX Chemotherapy with or without Aspirin

The authors investigate several players in the apoptotic cascade (pro and anti-apoptotic markers) in CRC, with a special focus on the African population.

The authors have 2 main approaches- patients and a CRC rat model and obrain several interesting results.

However I have several major concerns, but I believe authors will be able to answer accordingly. The results are not well presented-dispersed through several pages (same figure), also lack N etc, making it very hard to read think and analyse the data- I propose several improvements to help the reader. Also the rational is not well explained or even justified – making it hard to follow the rational, also lacks subtitles of the results also making it very hard to follow so many data.

ABSTRACT

3rd sentence of the abstract – is confusing or misleading ….

“Various cellular processes are associated with evasion of apoptosis. These include overexpression of pro-apoptotic and anti-apoptotic proteins (including BIRC7/Livin, Bcl-2, p53, Annexin V, PD-L1, and DARC) and dysregulation of DNA mismatch repair proteins (including MSH2 and PMS2).” this confusing because

BIRC7/Livin, Bcl-2 are anti-Apoptotic; p53 – pro-apoptotic ; PDL-1- can induce apoptosis and DARC- XXX

AnexinV is a marker not really a pro-apoptotic protein

So everything is put in the same bag and becomes very confusing..and many time the rational is lost.

So please re-write this sentence.

INTRODUCTION

Again concepts are mixed and confusing -please clarify .

1. I am not an expert on the apoptotic pathway etc but what I read was that BIRC7/Livin is an inhibitor of caspases – an anti-apoptotic protein and in the following sentences is stated this but also the opposite…so I don’t understand…:

Please clarify

“Thus, BIRC7/Livin serves as executioner of caspase-3 and-7 and also as an initiator of caspase-9 via the BIR domain (16). The BIR domain interacts with caspases active site to promote the degradation of active caspases (17). Apoptosis occurs when proteolytic activation of the cysteine-dependent 3 aspartate-directed proteases family (caspases) initiates and cleaves to effector caspases for the breakdown of intracellular protein substrates (18). BIRC7/Livin interacts with the second mitochondria-derived activator of caspases (SMAC) through the BIR domain to promote the caspase

activation in the cytochrome c pathway (13, 16)”.

2. Also needs revision:

The p53 protein is important for initiation of the apoptotic stimuli during anticancer therapy by

cleaving to the sites of damage in the cellular DNA (22). Thus, under normal cellular conditions, the p53 transcription factor regulates various cell functions such as, apoptosis activation and control of cell growth, migration, and invasion (23

P53 to my knowledge is able sense DNA damage and activate a series of cellular processes such as arrest cell cycle and induce apoptosis… not “cleaving to the sites of damage in the cellular DNA”

3. Also the role for AnexinV is not clear – Anexin5 can be used as a marker to detect apoptotic cells by its ability to bind to phosphatidylserine (outer leaflet of the plasma membrane). The protein per se, I believe the function is not yet very clear and has been highly correlated with coagulation and platelets – so I do wonder if the upregulation detected in the experiments Is not due to the TME and some platelet reaction…I think the authors should explain better the state of the art what is known and the different hypothesis..

4. Aspirin rational is not fully explained …confusing sentences… to my knowledge Aspirin by inhibiting COX can lead to an innate immune de-repression

“Aspirin inhibits cyclooxygenase (COX) enzymes (COX-1 and COX-2) -which are upregulated at sites of inflammation, to facilitate apoptosis (37).” Who facilitates apoptosis – aspirin? Or COX?

To my knowledge aspirin may work by inhibiting COX, reducing PGE2 production which induces a pro-tumor inflammatory profile and aspirin can revert this towards a classic anti-cancer immune pathway – therefore aspriirn could be used as adjuvant for immune checkpoint therapy (PMC4597191; DOI: 10.1016/j.cell.2015.08.015)

RESULTS

1. Figures are very difficult to follow with panels from the same figure spanning several pages ! Please put all panels in one figure A4 with the corresponding legend.

2. I could not find the N- number of patients analysed, number of models or number of independent experiments in all figures

3. Please put all pictures with the same magnification – also allows a better reading and analysis of the results

4. When possible please do dot charts – each dot a patient/ or mouse so we can analyse the dispersion and also number of data points

5. Please introduce subtitles of results

6. Could not find Table 3

7. It would be very important to show in your patient data – the correlation between the expression patern of the different markers and response to the neoadjuvant treatments – to understand for instance in Fig 3 the increase of BIRC7 in mucinous adenocarcinoma group upon FOLFOX treatment occurred in the ones that responded well or the poor responders? Within the good responders – they still have residual disease? Is this upregulation real or is just the escapers resistant clones that express high antiapoptotic proteins? the N is not specified …

8. Not clear when is adjuvant or neoadjuvant FO treatment

9. Fig. 6 why is FAcid only shown in I – also no comment why FA can induce so many effects…

10. Fig7 – results confusing sometimes you use anexin as a readout of apoptosis others not – but if you are using an antibody against Anexin is not the same thing as using anexin5 staining …confusing please clarify

11. Also all data of the rat CRC – could not see the impact of treatment on tumor progression – maybe I missed but it would be very important to correlate all results to response to treatment

DISCUSSION

“Cancer chemotherapeutics such as FOLFOX may induce apoptosis by targeting anti

apoptotic genetic mutations and equally reduce drug sensitivity which may lead to treatment failure

(38, 46).”

To my knowledge FOLFOX CT is thought to induce direct damage to cancer cells – DNA damage /cytotic damage etc that then induces apoptosis not targeting “apoptotic genetic mutations…”

6. PLOS authors have the option to publish the peer review history of their article (what does this mean?). If published, this will include your full peer review and any attached files.

Reviewer #1: No

Reviewer #2: No

---

## [Author Response · Author response to Decision Letter 0]

5 Nov 2020

Rebuttal letter to the Editor that responds to each point raised by the academic editor and reviewers:

Addressing points raised by the Editor: 

1- The article is here edited to PLOS ONE's style requirements.

2- Additional information about the patient records used in in the study, including as regard date range (month and year) during which patients' medical records were accessed and b) the date range (month and year) during which patients whose medical records were selected for this study sought treatment were addressed as follows:

 The series included 23 mucinous CRC and 69 CRC NOS FFPE tissue blocks selected from January 2009 to December 2017. Among these cases, 16 patients received neoadjuvant FOLFOX chemotherapy from September 2010 to December 2013 after a biopsy-confirmed histological diagnosis of CRC

3- Regarding animal care as per editorial guidelines, the following statements were added in the appropriate place and clarified: 

To minimize the rats suffering and distress, we worked as a team with veterinarians and animal care personnel from the ethic committee and staff members who are experienced in handling laboratory animals at all stage of the experiment and throughout the study period for appropriate monitoring and guidance.

The rats were purchased from Animal facility of the Faculty of Pharmaceutical Sciences, Ahmadu Bello University Zaria Nigeria.

The ethics committee at ABU (ABU committee on Animal use and care- https://abu.edu.ng/animal-use/) who reviewed and approved this study contains animal welfare experts from Faculty of Veterinary Medicine of the University. 

The collection of the blood was done by animal care expert under the supervision of senior veterinarian while restraining the rat manually and minimising the time of restrain and amount of blood collected so as to reduce stress and pain to the rats.

4- The word “Caucasian” was updated to “Whites” in the publication.

5- In the Methods section, the source, product number and lot numbers of the N-Methyl-N-Nitrosourea (NMU) used in the animal experiments in our study uodated as follows: 

N-Methyl-N-Nitrosourea (NMU; Shijiazhuang Aopharm Medical Technology Co., Ltd, China #684-93-5)

6- The source, product number/ lot numbers of the primary antibodies used in the histology analysis for our study were all provided as appropriate in the manuscript, see below: 

Anti-BIRC7 polyclonal antibodies from Antibodies-online (Aachen, Germany; ABIN358607; 1:80 dilution and ABIN672561; 1:100), Anti-Annexin V polyclonal antibody from Antibodies-online (Aachen Germany; ABIN4964891; 1:80 dilution), Anti-PD-L1 (CD274) monoclonal antibody from Antibodies-online (Aachen, Germany; ABIN5027498; 5 μg/mL), Anti-DARC polyclonal antibody from Antibodies-online (Aachen Germany; ABIN2821184; 1:50), Anti-MSH2 polyclonal antibody from Antibodies-online (Aachen Germany; ABIN3185692; 1:100), Anti-PMS2 polyclonal antibody from Antibodies-online (Aachen Germany; ABIN5546942; 1;30), Anti-Bcl-2 monoclonal antibody from Genemed Biotechnologies, Inc. (CA, USA; Clone Bcl-2-100; 1:60 dilution) and Anti-p53 monoclonal antibody from Genemed Biotechnologies, Inc. (CA, USA; Clone BP-53-12; 1:60 dilution). 

Anti-BIRC7 antibody from Antibodies-online (Aachen, Germany, ABIN672561; 1:100 dilution)

7- Additional information regarding statistical analysis were updated in the method section 

8- Scale bars were update in the microscopy and reflected in the figure legend. 

9- Tables were included as part of the manuscript

Addressing points raised by Reviewer 1: 

The 92 FFPE CRC patients samples used have only 16 patients treated with neoadjuvant chemotherapy as indicated in the method. These patients were not treated Aspirin. Only animal study was subjected to Aspirin. However, we envisage that there will be room to go ahead with human trial with Aspirin plus FOLFOX in future in African patients. Limited funding for translational research in Africa and the smaller sample size with limited number of patients who had received chemotherapy makes it difficult to analyse the relation between MSI, tumour stage and other biomarkers. However, we look forward to expanding the sample size to give room for these analyses when funding becomes available. Other parameters such as age location were updated in the manuscript.

Addressing points raised by Reviewer 2:

The sentence in the third line of the abstract was re-written as follows:

These include overexpression of pro-apoptotic and anti-apoptotic proteins (including p53 and PD-L1; BIRC7/Livin and Bcl-2), chemokine receptors (including DARC), and dysregulation of DNA mismatch repair proteins (including MSH2 and PMS2).

The points raised in introduction were revised as follows: 

1- Thus, caspase-3 -7 and -9 and the second mitochondria-derived activator of caspases (SMAC/DIABLO) are crucial interacting partners of BIRC7/Livin (16)

2- The p53 protein is important for initiation of the apoptotic stimuli during anticancer therapy by sensing DNA damage and activating a series of cellular processes.

3- Annexin V, an important marker for detection apoptotic cells by its ability to bind to phosphatidylserine (outer leaflet of the plasma membrane), has been reported to stimulate immunogenicity of tumor cells (25-26).

4- Aspirin inhibits cyclooxygenase (COX), reducing PGE2 production and inducing a pro-tumour inflammatory profile -Aspirin may revert this towards an important anti-cancer immune pathway and possibly serve as an adjuvant for immune checkpoint therapy [36-37].

All points raised by the reviewer in the result section were addressed accordingly. The numbers (N) were indicated in the figures appropriately.

Table 2, 3 and 4 were all placed appropriately for easy identification.

The small sample size for the human CRC treated for neoadjuvant FOLFOX limit our capacity to explore the reason behind expression pattern of the proteins in human subjects. However, we will make to increase sample size in future studies. 

We are of the view that treatment with Folinic acid alone without the addition of Oxaliplatin and 5-FU may have increase toxicity to cells generally. 

All other points raised were addressed. 

In the discussion, the sentence was re-written as follows: 

Cancer chemotherapeutics such as FOLFOX may induce direct damage to cancer cells with increase apoptosis and equally reduce drug sensitivity which may lead to treatment failure.

---

## [Decision Letter · Decision Letter 1]

14 Dec 2020

PONE-D-20-18593R1

Prognostic Significance of BIRC7/Livin, Bcl-2, p53, Annexin V, PD-L1, DARC, MSH2 and PMS2 in Colorectal Cancer Treated with FOLFOX Chemotherapy with or without Aspirin

PLOS ONE

Dear Dr. Faruk,

Thank you for submitting your manuscript to PLOS ONE. After careful consideration, we feel that it has merit but does not fully meet PLOS ONE’s publication criteria as it currently stands. Therefore, we invite you to submit a revised version of the manuscript that addresses the points raised during the review process.

Please address the issues indicated by the Reviewer 2.

We look forward to receiving your revised manuscript.

Kind regards,

Irina V. Lebedeva, Ph.D.

Academic Editor

PLOS ONE

Reviewers' comments:

Reviewer's Responses to Questions

**Comments to the Author**

1. If the authors have adequately addressed your comments raised in a previous round of review and you feel that this manuscript is now acceptable for publication, you may indicate that here to bypass the “Comments to the Author” section, enter your conflict of interest statement in the “Confidential to Editor” section, and submit your "Accept" recommendation.

Reviewer #1: All comments have been addressed

Reviewer #2: (No Response)

2. Is the manuscript technically sound, and do the data support the conclusions?

Reviewer #1: Yes

Reviewer #2: Yes

3. Has the statistical analysis been performed appropriately and rigorously? 

Reviewer #1: Yes

Reviewer #2: I Don't Know

4. Have the authors made all data underlying the findings in their manuscript fully available?

Reviewer #1: Yes

Reviewer #2: Yes

5. Is the manuscript presented in an intelligible fashion and written in standard English?

Reviewer #1: Yes

Reviewer #2: Yes

6. Review Comments to the Author

Reviewer #1: Well done! I think the authors have been diligent and presented a comprehensive range of data supporting their claim - which is an important one clincally

Reviewer #2: Overall I do not think the authors answer to all my concerns...

1-The sentence in the third line of the abstract was re-written as follows:

These include overexpression of pro-apoptotic and anti-apoptotic proteins (including p53 and PD-L1; BIRC7/Livin and Bcl-2), chemokine receptors (including DARC), and dysregulation of DNA mismatch repair proteins (including MSH2 and PMS2).

This continues confusing should put in separated brackets each class of genes

2- In the discussion, the sentence was re-written as follows:

Cancer chemotherapeutics such as FOLFOX may induce direct damage to cancer cells with increase apoptosis and equally reduce drug sensitivity which may lead to treatment failure.

Please clarify -what do the authors want to say? What do you mean by FOLFOX reducing drug sensitivity? That after FOLFOX treatment resistant clones emerge?

3- Could not find a clarification of why the rational for testing aspirin in the first place-was one of my concerns and i could not find where it was answered.

4-The figures continue in a very strange format - do not understand why Fig3 -is dispersed in 4 slides! fig4 in 3 slides???

5- no dot plots as requested

7. PLOS authors have the option to publish the peer review history of their article (what does this mean?). If published, this will include your full peer review and any attached files.

Reviewer #1: **Yes: **Finlay Macrae

Reviewer #2: No

---

## [Author Response · Author response to Decision Letter 1]

20 Dec 2020

Addressing points raised by Reviewer #2: 

1-The sentence in the third line of the abstract was re-written for second time as follows based on reviewers guide and suggestion to separate each class of gene(s) by bracket: 

Various cellular processes are associated with evasion of apoptosis. These include overexpression of pro-apoptotic protiens (including p53 and PD-L1), anti-apoptotic proteins (BIRC7/Livin and Bcl-2), chemokine receptors (including DARC), and dysregulation of DNA mismatch repair proteins (including MSH2 and PMS2).

2- In the discussion, the sentence was re-written for second time to address reviewer’s point as follows (This clarify our what we want to say):

Cancer chemotherapy regimens such as FOLFOX have been shown to significantly increased treatment efficacy and improved survival especially in metastatic CRC however, this treatment modality may be accompanied by complications such as myelotoxicity, neurotoxicity, non-alcoholic fatty liver disease and sinusoid obstruction syndrome (46).

3- The rational for testing aspirin in the first place is clearly indicated as per reviewer’s suggestions and for clarity as follows: 

Aspirin has been shown to be effective in preventing recurrence, decreasing risk of metastasis post curative therapy and primary prevention of CRC (47). However, complications from Aspirin use such as occult gastrointestinal bleeding, epistaxis extra and intra-cranial bleeding which are very rare, may resolve without much ramification (48). There is limited research that explores the mechanism through which aspirin alone and in combination with other chemotherapeutics exert anti-cancer effect especially in Black Africans with CRC. Thus, this strengthen the rational for testing Aspirin plus FOLFOX towards identifying biomarkers and clinical characteristics which may predict the benefit of Aspirin use for targeted invention in CRC management.

4-The figures are well re-arranged as suggested. 

5- Dot plots were presented as advised. 

I am attaching herewith our manuscript for you kind consideration.

---

## [Decision Letter · Decision Letter 2]

5 Jan 2021

Prognostic Significance of BIRC7/Livin, Bcl-2, p53, Annexin V, PD-L1, DARC, MSH2 and PMS2 in Colorectal Cancer Treated with FOLFOX Chemotherapy with or without Aspirin

PONE-D-20-18593R2

Dear Dr. Faruk,

We’re pleased to inform you that your manuscript has been judged scientifically suitable for publication and will be formally accepted for publication once it meets all outstanding technical requirements.

Kind regards,

Irina V. Lebedeva, Ph.D.

Academic Editor

PLOS ONE

Additional Editor Comments (optional):

Reviewers' comments:

Reviewer's Responses to Questions

**Comments to the Author**

1. If the authors have adequately addressed your comments raised in a previous round of review and you feel that this manuscript is now acceptable for publication, you may indicate that here to bypass the “Comments to the Author” section, enter your conflict of interest statement in the “Confidential to Editor” section, and submit your "Accept" recommendation.

Reviewer #2: All comments have been addressed

2. Is the manuscript technically sound, and do the data support the conclusions?

Reviewer #2: Yes

3. Has the statistical analysis been performed appropriately and rigorously? 

Reviewer #2: I Don't Know

4. Have the authors made all data underlying the findings in their manuscript fully available?

Reviewer #2: Yes

5. Is the manuscript presented in an intelligible fashion and written in standard English?

Reviewer #2: Yes

6. Review Comments to the Author

Reviewer #2: (No Response)

7. PLOS authors have the option to publish the peer review history of their article (what does this mean?). If published, this will include your full peer review and any attached files.

Reviewer #2: No

---

## [Editor Report · Acceptance letter]

7 Jan 2021

PONE-D-20-18593R2 

Prognostic Significance of BIRC7/Livin, Bcl-2, p53, Annexin V, PD-L1, DARC, MSH2 and PMS2 in Colorectal Cancer Treated with FOLFOX Chemotherapy with or without Aspirin 

Dear Dr. Faruk:

I'm pleased to inform you that your manuscript has been deemed suitable for publication in PLOS ONE. Congratulations! Your manuscript is now with our production department. 

Kind regards, 

on behalf of

Dr. Irina V. Lebedeva 

Academic Editor

PLOS ONE